# Transcriptomic and biochemical insights into key gene networks driving bulbil development of *Pinellia ternata* (Thunb.) Breit

Xiwei Jia[1,2☉], Xijia Jiu[1☉], Yuan Liu[1], Chao Guo[2], Dong Liu[1], Xin Zhao[1], Honggang Chen[1,3], Tao Du[1,3]*

**1** College of Pharmacy, Gansu University of Chinese Medicine, Lanzhou, Gansu Province, China, **2** College of Life Sciences and Technology, Ningxia Polytechnic, Yinchuan, Ningxia Hui Autonomous Region, China, **3** Northwest Chinese and Tibetan Medicine Collaborative Innovation Center, Gansu University of Chinese Medicine, Lanzhou, Gansu Province, China

☉ These authors contributed equally to this work.
\* gslzdt@163.com

**Data Availability Statement:** All relevant data are contained in this paper at NCBI SRA: PRJNA903650 and in the Figshare database,

## Abstract

In this study, we explored the developmental characteristics of *Pinellia ternate* bulbils as well as the key gene networks driving the development of bulbils. Based on physiological and biochemical reactions as well as transcriptome technology, this study determined the content of endogenous metabolites and related enzyme activities during the five growth stages of the bulbils, obtained the transcriptome information of all samples. The results showed that the contents of sucrose and starch increased significantly in the ZY_2 and ZY_4 stages, and the changes in the activities of SPS, SuSy, and SS were basically consistent with the changing characteristics of sucrose and starch content. The contents of ABA and JA generally showed an increasing trend from ZY_1 to ZY_4, while the content of IAA was significantly higher only in ZY_1 and ZY_4 stages compared to other stages. In order to get more bioinformatic support for these results, RNA-Seq analysis was performed. There were 12 key enzyme genes differentially expressed in the sucrose-starch metabolic pathway, and 14 enzyme genes differentially expressed in the above-mentioned endogenous hormone metabolic pathway. Their expression characteristics well supported the measurement results of physiological and biochemical substances. Our results showed that ZY_2 and ZY_4 stages are the critical periods for the accumulation of sucrose and starch in the bulbils. JA has an important role in the whole development process of bulbils, which may enhance the adaptability of the bulbils to the environment in the transition process from the tender to the mature tissues. The low concentration of GA was beneficial to the normal development of the bulbils. IAA may have a strong regulatory role in the initial formation stage of the bulbils, which is beneficial to their tissue differentiation. In addition, four core transcripts involved in the bulbils development process were screened using WGCNA. This study provides an information source for analyzing the molecular mechanism of bulbils growth and development, and also helps to address the lack of genetic information in non-model plant species.

dataset DOI number: 10.6084/m9.figshare.
27021268.

**Funding:** This work was supported by the Science
and Technology Key R&D Program in Gansu
Province (21YF5NA130) and the Special
Foundation for Construction of National Traditional
Chinese Medicine Industry Technology System in
China "Supported by the earmarked fund for CARS-
21".There was no additional external funding
received for this study. The funders had no role in
study design, data collection and analysis, decision
to publish, or preparation of the manuscript.

**Competing interests:** The authors declare there are
no competing interests.

## Introduction

Bulbils are special organs for asexual reproduction formed by some plants in nature [1]. Some
of these Bulbils are formed in the axils of the plant's leaves and thus grow above ground, while
others are formed underground, which share similarities with tubers, bulbs and bulblets.
Plants such as the genus Pinellia (family Araceae), the genus Lilium (family Liliaceae), the
genus Sedum (family Crassulaceae), and the genus Dioscorea (family Dioscoreaceae), produce
bulbils containing a lot of nutrients and secondary metabolites [2]. However, the bulbil of
*Pinellia ternate* is a typical representative of these bulbils, the bulbils are not only propagation
material, but can also be used as medicine after processing and drying [3]. Currently, *P. ternata*
serves as a botanical medicine commonly used by traditional Chinese medicine practitioners
in China and many Southeast Asian countries [4]. According to the theory of traditional Chi-
nese medicine, *P. ternata* can be applied to eliminate dampness to reduce phlegm, descend
perverse rise of stomach-qi to stop vomiting, dissolve lumps and resolve masses [5, 6]. How-
ever, the chemical constituents of *P. ternata* are relatively complex, including organic acids,
proteins, amino acids, alkaloids, polysaccharides, and trace elements, among which organic
acids ehibit antitussive, expectorant and anti-tumor properties [7]. As reported by modern
pharmacological studies, *P. ternata* can relieve cough, dissipate phlegm, and stop vomiting,
and exert anti-gastric ulcer, coagulant, anti-tumor, anti-bacterial, anti-inflammatory, anti-oxi-
dative, anti-epileptic and other pharmacological effects, acting as a prospect for medicine
development [8, 9]. Since *P. ternata* has been proven to be effective, there is a growing demand
for it in China and foreign countries, and the wild medicinal resources can no longer meet the
market demand [10]. Notably, *P. ternata* is a vital herb in the traditional Chinese medicine for-
mula for treating and preventing COVID-19 [11, 12]. According to statistics, in the first half of
2020, the total export volume of *P. ternata* exhibited a year-on-year growth of about 34% [13].

 *P. ternata* bulbils have become the focus of production workers during operations because
of their dual roles as medicinal materials and propagation materials. A large number of studies
have been conducted on the bulbils of *P. ternata*, including the development of bulbils, the
mechanism of bulbil genesis, the response of bulbils to biotic or abiotic stresses, and the high-
yield cultivation techniques of bulbils, and so on. Similar to other bulbous species, the develop-
ment of bulbils is closely related to phytohormones, and there is evidence that Indole-3-acetic
acid (IAA) synthesized by the leaves and transmitted by the petiole promotes the initiation of
bulbils and the early development of bulbils in *P. ternate* [14]. In some plants such as lilies,
IAA may act on bulbil genesis and development either indirectly or in synergy with cytokinins
(CKs) and strigolactone (SL) [15]. In addition, several growth hormone analogs have shown
modulation of bulbils [16]. The complex regulation of IAA on the bulbils of *P. ternata* has not
been deeply revealed yet. Brassinolide (BR) and CK have long been recognized to play impor-
tant roles in asexual reproduction, and the release of CYCD and ARF proteins triggered by
BRs in *P. ternate* bulbils further regulates the expression of a series of downstream genes [17].
CKs, on the other hand, are known to play a role in the early initiation of bulbils, and these
effects may also interact with other hormones. This has been corroborated in species such as
*Lilium lancifolium* [18] and *Lycoris chinensis* [19]. Unlike CKs, the regulatory effects of GA are
notable in the later stages of the development of *P. ternata* bulbils [20]. Commonly considered
a developmental inhibitor, abscisic acid (ABA) inhibits stem elongation and bulbil sprouting
in *P. ternata* and promotes bulbil dormancy [21]. A recent study suggests that salicylic acid
(SA), and jasmonic acid (JA) may act synergistically to positively regulate the genesis of *P. ter-
nata* bulbils [22]. It has also been reported that cytokinin (CK) promotes potato tuber forma-
tion, whereas gibberellin (GA) and strigolactone (SL) exert inhibitory effects [23]. CK
positively regulates lily bulblets formation [18]. It is well known that sugar metabolism plays

the role of the most basic energy conversion as well as storage in the plant body. The development of bulbils in *P. ternata* is essentially a transfer of energy substances, and the catabolic transfer of sucrose and starch is the starting point of bud development [24]. In addition, the role of sugar metabolism for bulbils may not be limited to initiation, as it has been shown that a large accumulation of starch is observed in the later stages of development of hemlock bulbils [25]. Similar findings have been made in lily bulbs [26]. Previous studies have explored sucrose and starch metabolism in *Sagittaria sagittifolia* bulbils using transcriptomics and identified important genes in the metabolic pathways, indicating that sucrose and starch metabolism play crucial roles in bulbils formation and development [27]. In higher plants, sucrose and starch are the main forms of carbohydrate transport and storage, respectively, and starch synthesis requires the catabolic conversion of sucrose in the plant's "storage" organs [28]. The metabolism of sucrose and starch is very complex, involving dozens of enzymes, and has therefore attracted the attention of many scholars in the field of plant physiology and biochemistry [29, 30]. In addition, sucrose and starch metabolism are major metabolic pathways during lily bulb growth and development, in which the expression levels of sucrose synthase (SuSy) and invertase (INV) are significantly up-regulated at the stage of bulb scale differentiation, and sucrose phosphate synthase (SPS) is significantly up-regulated at the stage of bulb morphogenesis. Meanwhile, the expression levels of enzymes involved in starch synthesis differed between the different growth stages of bulblets and mother bulbs [31]. Another study showed that sucrose level may be the main regulator of potato tuber formation, and the formation of tubers in vitro displays high dependence on sucrose concentration [32].

Endogenous hormones and sugar metabolism, as the two core modules of *P. ternata* bulbil sprouting, have been the subject of numerous previous studies. However, the large number of gene regulatory networks involved and the regulatory relationships among these genes are the trend and direction of future research on *P. ternata* bulbils. Therefore, In this study, according to the shape of bulbils, their growth was divided into five stages, namely, initial formation stage (ZY_1), expansion stage (ZY_2), mature stage (ZY_3), secondary seedling emergence period (ZY_4) and harvest stage (ZY_5). The physiological and biochemical indexes of the bulbil samples obtained at the five stages were detected, and then the transcriptome sequencing was carried out. Next, the starch and sucrose metabolic pathways, endogenous hormone metabolic pathways, and transcription factor (TF) and hormone signal transduction pathways were analyzed. At last, based on the Weighted Gene Co-expression Network Analysis (WGCNA) algorithm, a whole transcriptome co-expression network was constructed for phenotypic traits, so as to screen potential transcripts related to endogenous hormone, sucrose and starch metabolic pathways. Through WGCNA, the transcriptional regulatory network of sugar metabolism and endogenous hormones affected by the development of bulbils was identified, which will help to discover functional genes and molecular mechanisms related to the development of *P. ternata* bulbils.

## Materials & methods

### Plant materials

Sample collection for this experiment was done at the planting bases of *P. ternata* in Qingshui County, Tianshui City, Gansu Province, China, and the subsequent experimental content was done at Gansu University of Traditional Chinese Medicine. *P. ternata* bulbils of the same size (0.8–1.0 cm) were planted in three plots (10 m² per plot). The propagation materials used for the study as well as the samples collected was *Pinellia ternata* (Thunb.) Breit. Which originated from the ecotype of Shandong Province, China. After planting, the growth state of bulbils was subjected to frequent observation. Samples were collected when new bulbils were initially formed (ZY_1, the first stage), tissues of nascent bulbils were obtained as samples by using a scalpel. And then

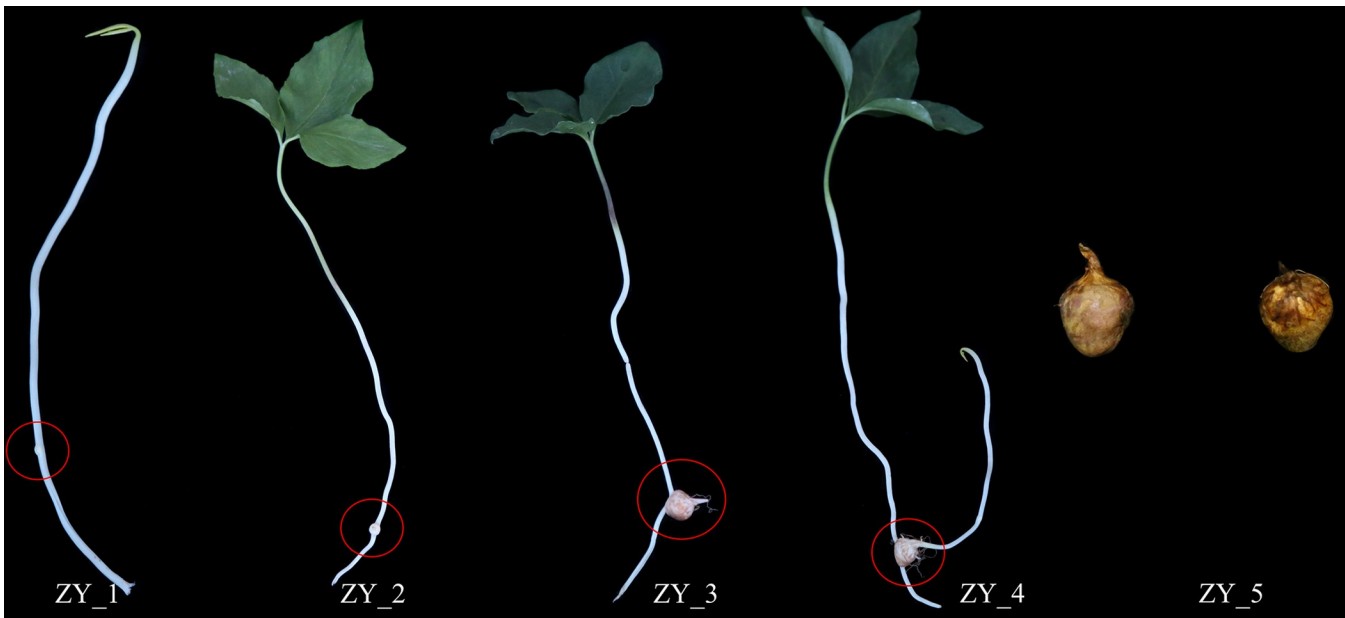

**Fig 1. Morphological characteristics of bulbils at different growth stages.**

collected at 26 d (ZY_2, expansion stage), 52 d (ZY_3, mature stage), 78 d (ZY_4, secondary seedling emergence period) and 104 d (ZY_5, harvest period) by the same method. Respectively, The development of bulbils is routinely divided into three stages: initiation stage, expansion stage, and mature stage. However, the growth and development characteristics of the *P. ternata* bulbils are different from those of other plants, in that after the bulbils have matured, they can sprout petioles again to form a new plant to continue growing. This process continues until the external temperature is not suitable for its growth, the aboveground part withered, at this time the bulbils are harvested, the growth cycle is complete. So we set up two additional sampling stages, ZY_4 and ZY_5, and the changes in the morphology of *P. ternata* bulbils at different sampling stages are shown in Fig 1. Immediately after the rinsing with distilled water, the samples were put into liquid nitrogen for quick freezing, and then stored in a -80˚C refrigerator until further analysis.

### Determination of the content of endogenous hormones, sucrose and starch as well as the activities of sucrose and starch synthesis-related enzymes

Enzyme-linked immuno sorbent assay was adopted to determine the content of endogenous hormones and the activities of sucrose and starch synthesis-related enzymes. Specifically, fresh samples (0.1 g) were extracted with 0.9 mL of phosphate-buffered saline (pH 7.4) and centrifuged at 1000×g for 20 min according to the instructions of IAA, GA, ABA, JA, SPS, SuSy and SS detection kits (Shanghai QCheng Biotechnology, China) [33, 34]. In addition, the kits provided by Nanjing Jiancheng Bioengineering Institute were utilized to determine the sucrose content and starch content by ultraviolet colorimetry and spectrophotometry, respectively, according to the kit instructions for specific steps.

### Ribonucleic acid (RNA) isolation, complementary deoxyribonucleic acid (cDNA) database preparation, and transcriptome sequencing

Briefly, according to the designated scheme, the total RNA was extracted from samples (RNA-prep Pure Plant Plus Kit; DP441), and the DNA was digested by deoxyribonuclease (DNase).

Then poly A-containing mRNA was enriched using oligo (dT)-attached magnetic beads, and then randomly fragmented into small segments. The first-strand cDNA and second-strand cDNA were synthesized with small fragments of the mRNA as templates. Next, after purification by Agencourt AMPure XP (BECKMAN COULTER, A63881), the double-stranded cDNA was subjected to terminal repair and poly (A) addition. Finally, sequencing adapters were ligated to the 50 and 30 ends of the fragments. The fragments were purified via agarose gel electrophoresis, and finally polymerase chain reaction (PCR) amplification. After qualified by Agilent 2100 Bioanalyzer, the constructed database was sequenced by Shanghai OE Biotech Co., Ltd. using Illumina HiSeq TM 2500.

## Bioinformatic analysis

Trimmomatic (Version: 0.36; Parameters: LEADING:3 TRAILING:3 SLIDINGWIN-DOW:4:15 MINLEN: 50) [35] was employed for raw data (raw reads) processing in FASTQ format, and clean reads were obtained after the removal of reads containing ploy-N and low-quality reads, and then they were assembled into expressed sequence tag clusters (contig), followed by de novo assembly into transcripts using Trinity software (Version: 2.4.0; Parameters: —seqType fq—SS_lib_type RF) [36]. Next, unigenes were aligned with NCBI's non-redundant (NR), Swiss-Prot, and gene phylogenetic databases to annotate their functions. After the functional annotation of unigenes, the number of reads aligned to unigenes in each sample was obtained using bowtie2 software(Version: 2.3.3.1; Parameters:—reorder -k30 -t) [37], and the expression level [the value of fragments per kilobase of exon model per million mapped fragments (FPKM)] of unigenes was calculated by eXpress software (Version: 1.5.1; Parameters:—rf-stranded) [38]. Differentially expressed genes (DEGs) were filtered with $q < 0.05$ and fold-Change $> 2$ as the criteria. Based on the hypergeometric distribution, Gene Ontology (GO) enrichment and Kyoto Encyclopedia of Genes and Genomes (KEGG) pathway enrichment analyses of DEGs were performed by R Bioconductor package (Version: 3.2.0), respectively.

## Quantitative reverse transcription and PCR (qRT-PCR)

QRT-PCR was used to verify 9 DEGs involved in the metabolic pathways and signal transduction pathways of starch, sucrose and endogenous hormones. QRT-PCR included two steps, namely, RT and PCR. Step one: RT reaction system was prepared using 0.5 μg of RNA, which was kept at 42˚C for 15 min on the ABI 970 PCR instrument, and then the RT enzyme and gDNA remover were inactivated at 85˚C for 5 s, and cooled at 4˚C. Next, the cDNA in each tube was diluted using 90 μL of nuclease-free $H_2O$ to obtain the final volume of 100 μL, and detected on the machine after the PCR Mix. The specific steps were 94˚C for 30s, 94˚C for 5s and 60˚C for 30s, 45 cycles. At the end of the cycle, the product specificity was detected using a melting curve: the temperature was slowly increased from 60˚C to 97˚C, and the fluorescence signal was collected five times per˚C. Finally, the relative expression levels of genes were normalized and calculated by $2^{-\Delta\Delta Ct}$ method. The pcr primer sequences and internal reference genes are shown in S2 Table.

## Data analysis

SPSS software (26.0, IBM, Chicago, United States) and Excel 2016 were adopted to statistically analyze all data. The physiological responses of bulbils and the differences in the levels of sucrose, starch and hormones at different growth stages were examined by analysis of variance (ANOVA). The least-significant difference (LSD) of significant data was calculated at $p < 0.05$, and the results were expressed as mean ± standard error.

## Results

### Variation characteristics of carbohydrate content and related enzyme activities in *P. ternata* bulbils at different growth stages

The contents of sucrose, starch and the activities of sucrose phosphate synthetase, sucrose synthetase and starch synthetase were measured at different growth stages. The results showed that the changes of sucrose content and starch content were basically the same during the growth period, the contents of both increased significantly in the ZY_2 and ZY_4 stages, and were significantly higher in the ZY_4 stage than in other stages, and the contents of both decreased significantly in the ZY_5 stage. The difference was that sucrose content decreased significantly in the ZY_3 stage, while starch content still increased slowly (Fig 2A and 2B). The characteristics of sucrose and starch content in the five stages are closely related to the growth law of *P. ternata*. The ZY_2 and ZY_4 stages are the key stages for polysaccharide accumulation in the bulbils. In the ZY_3 stage, the germination of new leaf stalks requires energy consumption and active physiological and biochemical reactions, so it is speculated that the

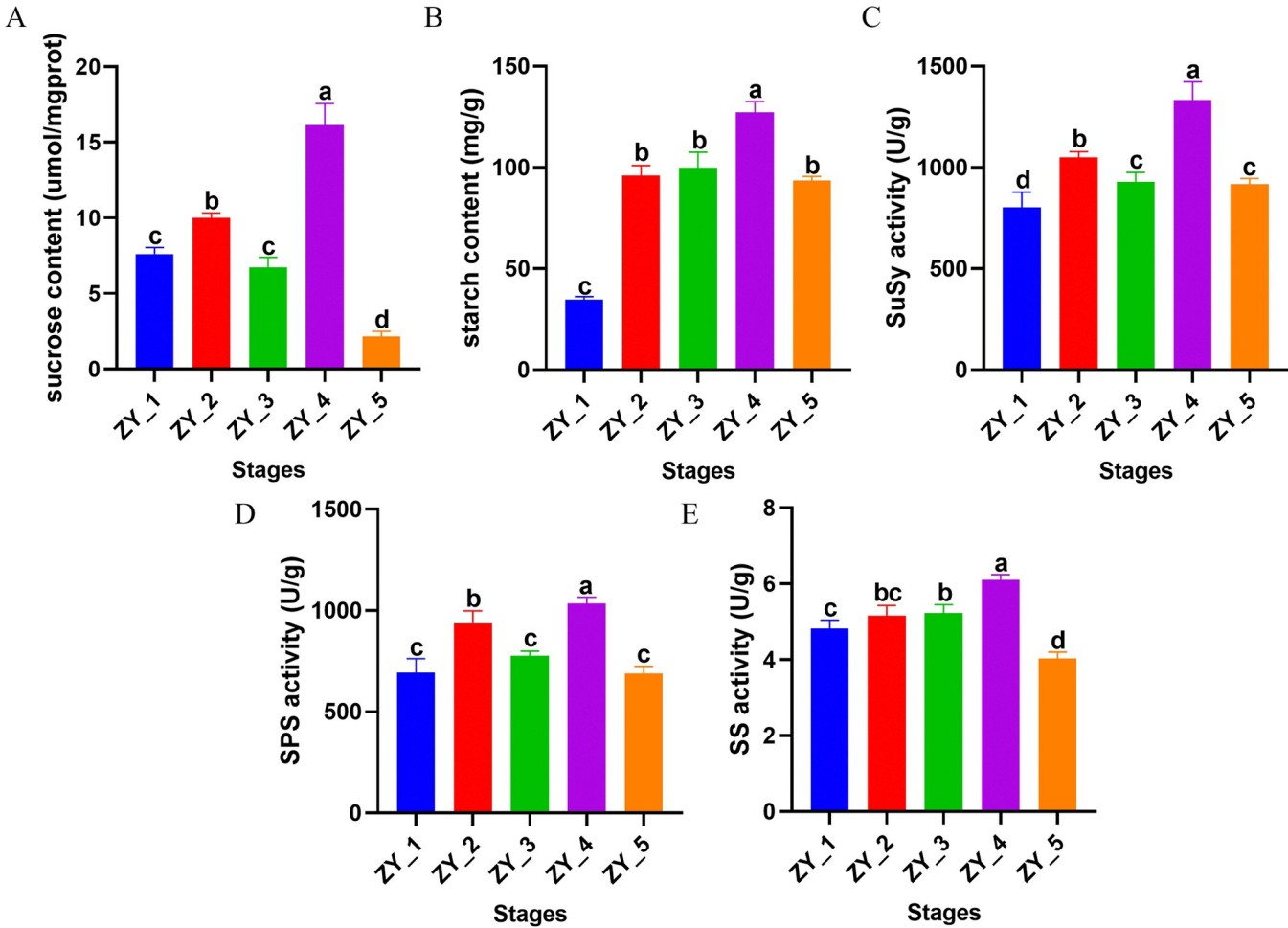

**Fig 2. Determination of carbohydrate content and related enzyme activities in *Pinellia ternata* bulbils at different growth stages.** A. Determination of sucrose content at different stages. B. Determination of starch content at different stages. C. Determination of SuSy activity at different stages. D. Determination of SPS activity at different stages. E. Determination of SS activity at different stages. The x-axis displays different growth stages, while the y-axis represents the content or activity of the measured substance.

decrease of sucrose content is closely related to this process. In the ZY_5 stage, the leaves and branches of *P. ternata* withered, and the content of sucrose and starch in the bulbils decreased. The change in characteristics of sucrose content were consistent with those of sucrose synthetase and sucrose phosphate synthetase (Fig 2C and 2D), however, the variation characteristics of starch synthase and starch content in bulbils were similar (Fig 2E).

## Variation characteristics of endogenous hormone content in *P. ternata* bulbils at different growth stages

The contents of endogenous hormones in *P. ternata* bulbils at 5 growth stages were measured. The results showed that there was no difference in gibberellin content between the ZY_1 and ZY_4 stages, which remained at a low level and decreased significantly in the ZY_5 stage (Fig 3A). The content of abscisic acid increased significantly only at the ZY_4 stage, and then decreased significantly at the ZY_5 stage (Fig 3B). Jasmonic acid content generally showed an increasing trend during the growth and development of bulbils, and reached the maximum value in the ZY_4 stage (Fig 3C). Auxin content generally decreased in the early stage, and significantly increased in the ZY_4 stage (Fig 3D). In general, the contents of the four hormones decreased significantly in the ZY_5 period, which was the end of *Pinellia ternate* growth period.

## Whole-genome analysis of *P. ternata* bulbils at different growth stages

The transcriptome data of *P. ternata* bulbils at five different growth stages were compared using the second-generation transcriptome sequencing method on the Illumina platform. Each group of samples had three biological replicates, totaling 15 samples, and a total of 107.62 G of Clean reads were obtained. The effective data volume of each sample ranged from 6.87 to 7.45 G, the Q30 base (Bases with Qphred values greater than 30 in raw bases) distribution was 90.18–91.83%, and the average GC content was 53.24%. 68,391 unigenes were obtained, with a total length of 69,797,270 bp and an average length of 1,020.52 bp. A total of 40,251 genes were annotated to the NR database, 26,966 genes to the Swissprot database, 8,084 genes to the KEGG database, 21,371 genes to the KOG database, 33,841 genes to the eggNOG database, 23,296 genes to the GO database, and 22,343 genes to the Pfam database (Fig 4A). Subsequently, the reads were aligned to unigenes, with an alignment rate of 80.56–88.67%, which enriched the available genomic information of *P. ternata*.

According to the statistical analysis of DEGs in *P. ternata* bulbils at different growth stages, totally 11,880 DEGs (7,472 up-regulated and 4,408 down-regulated) between ZY_2 and ZY_1, 9,865 DEGs (5,253 up-regulated and 4,612 down-regulated) between ZY_3 and ZY_2, 7,989 DEGs (3,606 up-regulated and 4,383 down-regulated) between ZY_4 and ZY_3, 24,785 DEGs (11,617 up-regulated and 13,168 down-regulated) between ZY_5 and ZY_4 were identified (Fig 4B). There were 857 DEGs shared by the four comparison groups of samples, which suggested that these 857 transcripts are continuously expressed at different growth and development stages of *P. ternata* bulbils (Fig 4C). The NR database was used for the functional annotation of *P. ternata* transcripts (Fig 4D). The results revealed that the number of annotations in *Colocasia esculenta* was the largest [29,399 (73.04%)], while that in *Macleaya cordata* was the smallest [246 (0.61%)]. In order to verify the reliability of RNA-Seq data, 9 single genes encoded, namely, GBE1, pgm, WAXY, E2.4.1.13, IAA, LUT5, ABF, E1.14.17.4 and AOS, were selected for qRT-PCR validation. The results manifested that there was a good agreement between RNA-Seq data and qRT-PCR data ($r^2 = 0.9753$), proving the reliability of the relevant gene expression levels (Fig 4E).

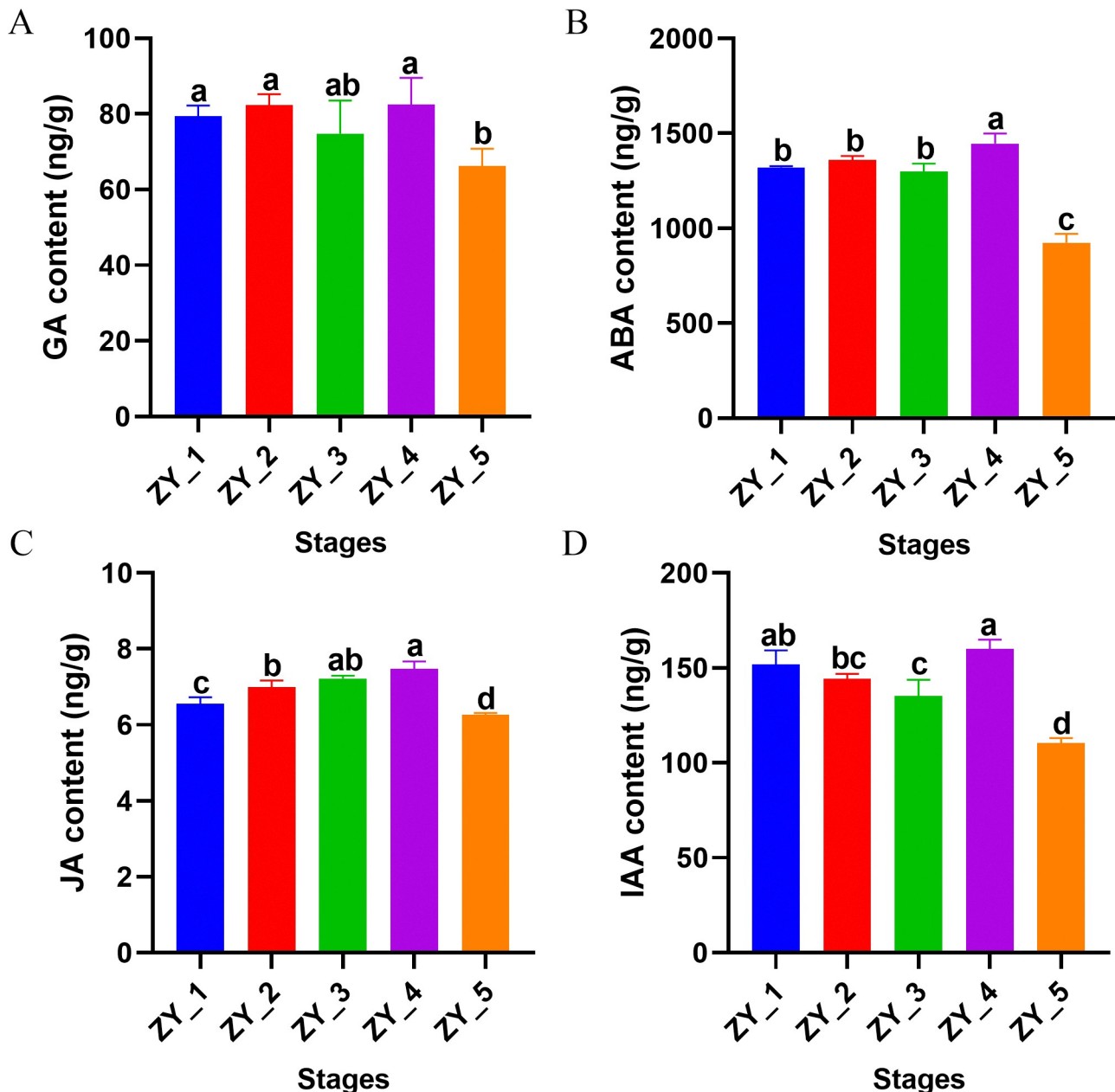

**Fig 3. Determination of endogenous hormone content in *Pinellia ternata* bulbils at different growth stages.** A. Determination of GA content at different stages. B. Determination of ABA content at different stages. C. Determination of JA content at different stages. D. Determination of IAA content at different stages. The x-axis represents different growth stages, while the y-axis represents the content of the measured substance.

## KEGG: Pathway enrichment analysis of DEGs

To uncover the dynamic characteristics of the growth and development of *P. ternata* bulbils at different stages, the transcripts of the bulbils at five stages were compared. KEGG analysis showed that the gene expression pathways and biosynthesis pathways of *P. ternata* bulbils differed at different growth stages. It was also found that 12 biological expression pathways, namely, photosynthesis (ko00195), photosynthetic-antenna proteins (ko00196), α-linolenic acid metabolism (ko00592), phenylpropanoid biosynthesis (ko00940), flavonoid biosynthesis

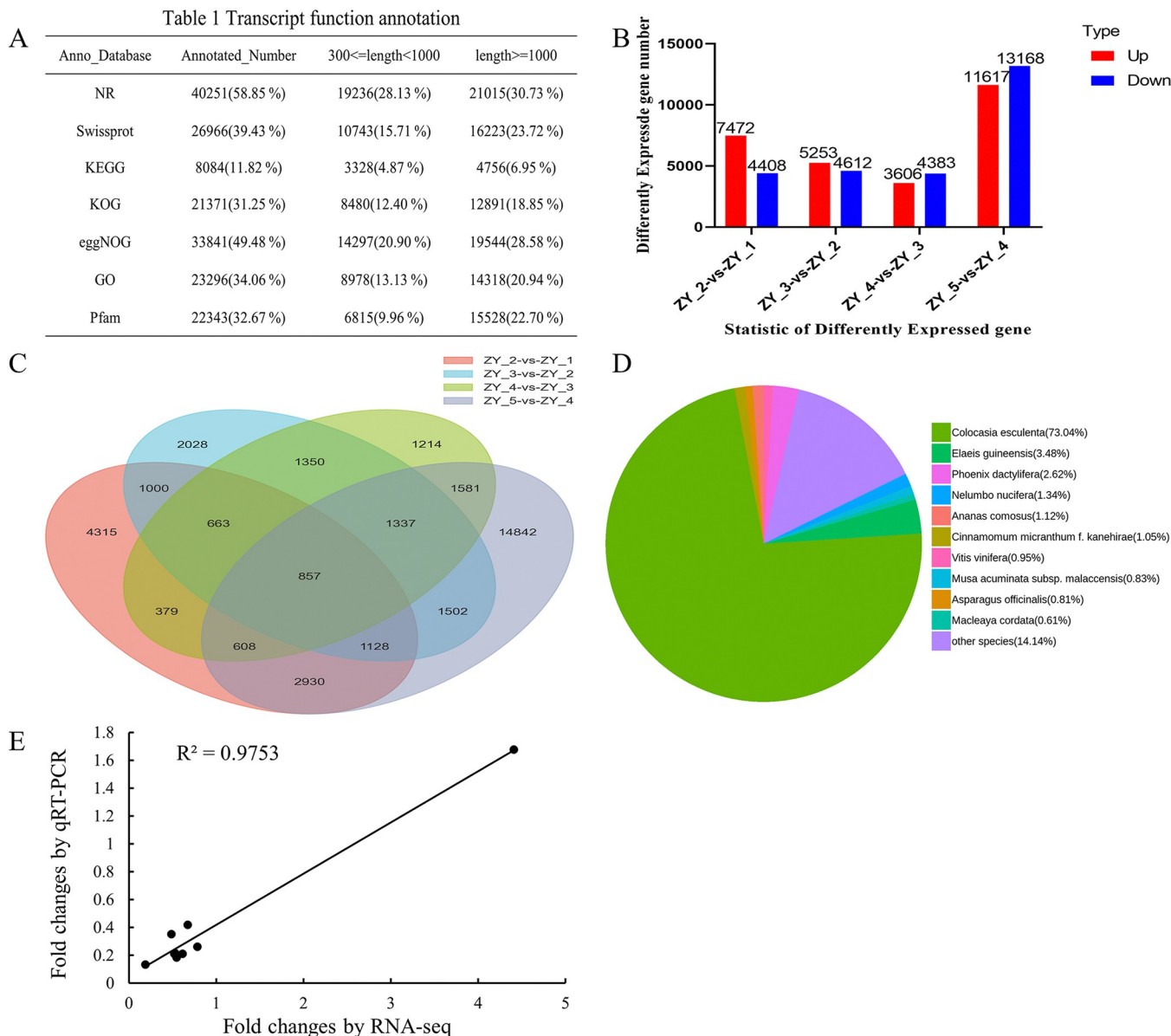

**Fig 4. Statistics of transcriptome sequencing results of *Pinellia ternata* bulbils at different growth stages on the Illumina platform.** A. The number of annotations of the spliced unigene sequences in the NR, Swiss-Prot, KEGG, KOG, eggNOG, GO, and Pfam databases. B. The number of DEGs in different comparison groups. C. Venn diagram of a single gene in each group. D. Species distribution of the identified transcripts in the NR database. E. Correlation analysis of the qRT-PCR and RNA sequencing (RNA-Seq) results for 9 genes.

(ko00941), plant hormone signal transduction (ko04075), pentose and glucuronic acid inter-conversion (ko00040), cutin (ko00073), suberin and wax biosynthesis, starch and sucrose metabolism (ko00500), biotin metabolism (ko00780), brassinosteroid biosynthesis (ko00905), and plant-pathogen interaction (ko04626) were evidently enriched in the bulbils, indicating that these 12 pathways play an important role in the whole-life process of bulbils including formation, growth and development. *P. ternata* bulbils, formed and attached to the underground petioles, are vital "reservoir" organs of plants in the soil responsible for storing nutrients. Photosynthesis exerts a crucial effect on the yield and quality of bulbils. Two biological pathways, photosynthesis (ko00195) and photosynthetic-antenna proteins (ko00196), were significantly

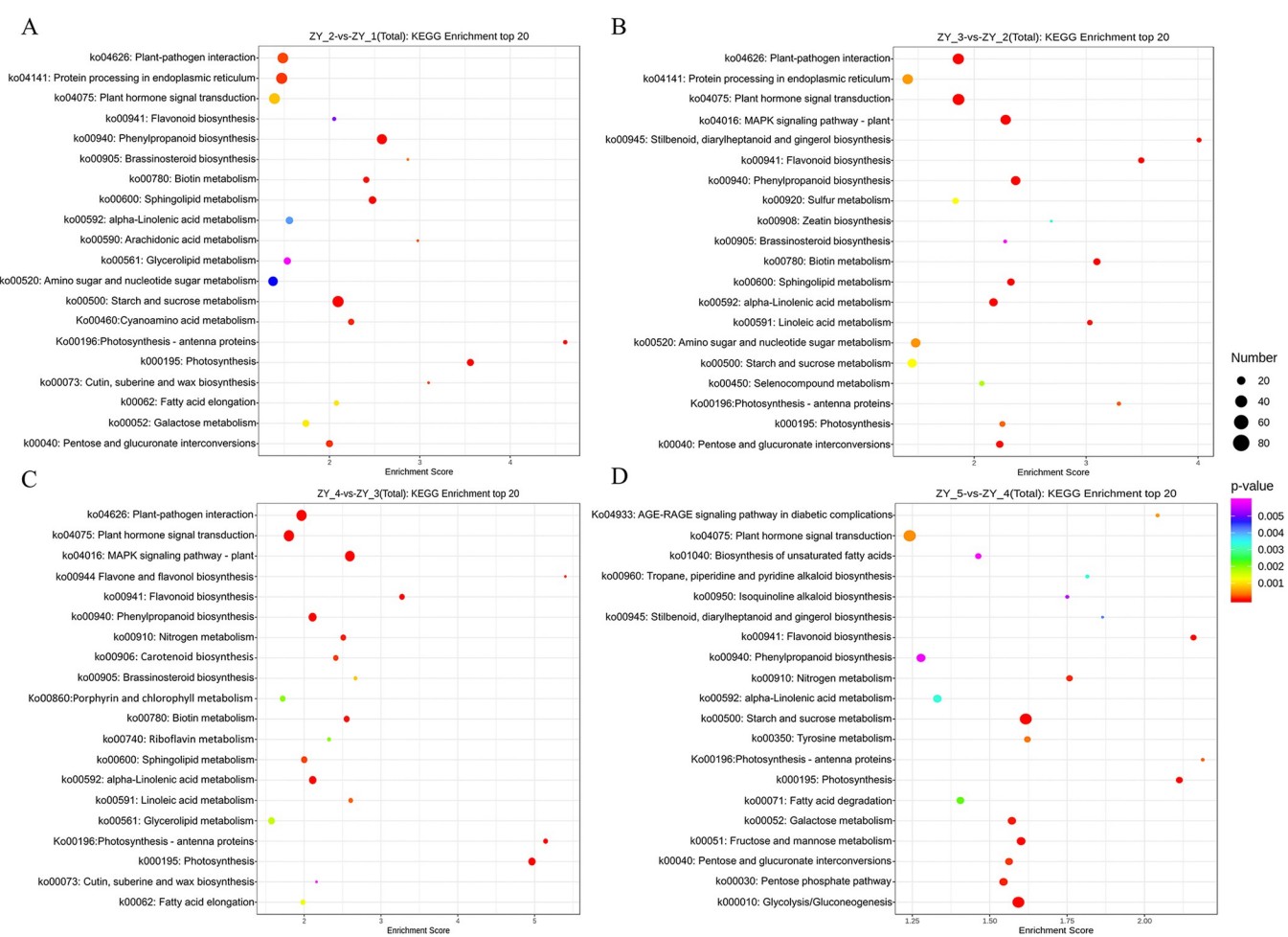

**Fig 5. KEGG enrichment analysis of gene expression pathways in *Pinellia ternata* bulbils at different growth stages.**

enriched in the bulbils at different growth stages, indicating that the bulbils may be a key site for the regulation of photosynthesis of plants, and antenna proteins are integral parts for light-harvesting complex to obtain light energy in photosynthesis. Remarkable enrichment of the biosynthetic pathways of cutin (ko00073), suberin, and wax could be observed at ZY_1, ZY_2 and ZY_3, which indicated that more wax and lignin are synthesized and accumulated in the bulbils during the immature period (Fig 5A and 5B). The enrichment results of pentose and glucuronic acid interconversion (ko00040), starch and sucrose metabolism (ko00500), α-lino-lenic acid metabolism, phenylpropanoid biosynthesis (ko00940), flavonoid biosynthesis (ko00941), plant hormone signal transduction (ko04075) and brassinosteroid biosynthesis (ko00905) demonstrated that primary and secondary metabolites were continuously synthe-sized and accumulated in bulbils at different growth stages (Fig 5A–5D).

## DEGs related to sucrose and starch metabolic pathways

*P. ternata* bulbils are "reservoir" organs, where sucrose and starch are their vital storage sub-stances, laying a solid material foundation in the morphogenesis process of the bulbils. For fur-ther exploration of the dynamic variation characteristics and metabolic mechanisms of sucrose and starch synthesis and accumulation in the bulbils at different growth and

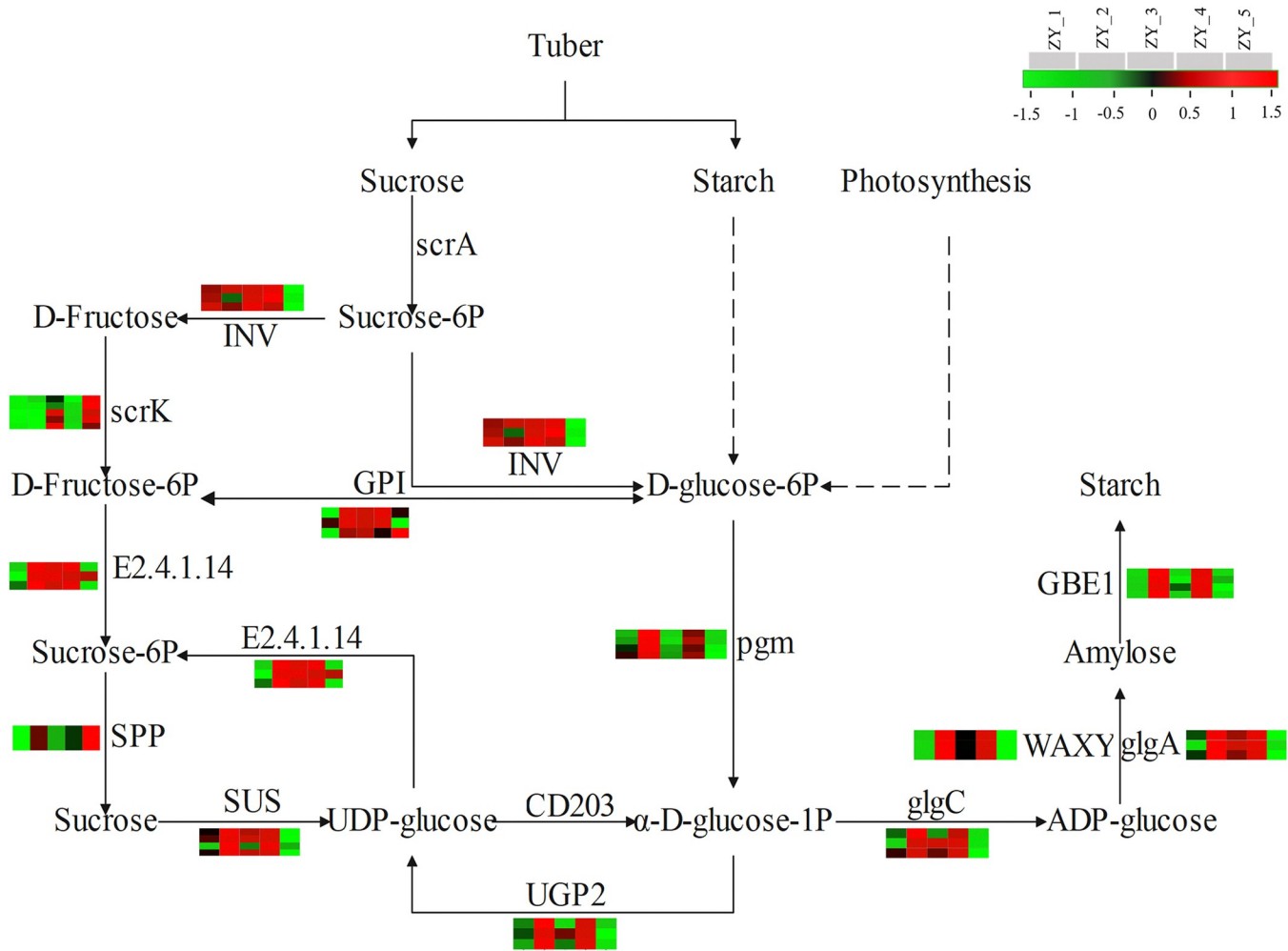

**Fig 6. Heat maps of DEGs related to sucrose and starch synthesis in the bulbils at different growth stages plotted by log2 transformation of FPKM values.**

development stages, their biological pathways and related gene expression levels were subjected to visualized analysis. According to KEGG data, there were 12 key DEGs in the sucrose and starch metabolic pathways in the bulbils at different growth stages. As illustrated in Fig 6, variations in the key DEGs related to sucrose and starch metabolism (INV, scrK, E2.4.1.14, SPP, SUS, UGP2, pgm, glgC, GPI, glgA, WAXY and GBE1), at different growth stages revealed that these genes were expressed at different levels in the heat maps at different stages. Except for the high expression levels of INV at ZY_1, ZY_2, ZY_3 and ZY_4, the expression levels of the other 11 genes were up-regulated at ZY_2, ZY_3 and ZY_4, especially at ZY_2 and ZY_4, indicating that sucrose and starch metabolism are more active at ZY_2 and ZY_4. In the present study, sucrose was the main substrate for the synthesis and accumulation of starch in the bulbils. Sucrose underwent a series of transformation and metabolism to produce uridine diphosphate glucose and D-glucose 6-phosphate. Then the same intermediate α-glucose-1p generated under the action of two key genes, UGP2 and pgm, was finally produced into starch under the action of key genes, glgC, glgA, WAXY and GBE1. The expression of these key enzyme genes was significantly up-regulated in ZY_2 and ZY_4 periods, which was consistent with the change of starch content.

## DEGs related to the endogenous hormone metabolic pathway

As trace and efficient organic compounds synthesized in plants, endogenous hormones are crucial for plant organogenesis, morphogenesis and other growth and development processes. For further exploration of the metabolic mechanisms and dynamic variation characteristics of endogenous hormones in *P. ternata* bulbils at different growth and development stages, the metabolic pathways of four endogenous hormones and the expression levels of related genes were examined. The heat maps (Fig 7) illustrated variations in ABA, GA, IAA, JA and other related enzymes in the metabolic pathways at different growth stages, and revealed that these genes were differentially expressed at different growth stages. According to the KEGG enrichment analysis results, it was found that four enzymes [15-cis phytoene synthase (crtB), ζ-carotene desaturase (ZDS), β-carotene hydroxylase (LUT5) and 9-cis-epoxycarotenoid dioxygenase (NCED)] encoded by eight highly expressed DEGs were involved in the metabolic pathway of ABA. In the first place, geranylgeranyl pyrophosphate was converted into prephytoene diphosphate and then into crtB, whose expression level was the highest at ZY_2 and ZY_4. Next, ZDS and NCED were involved in the production of lycopene and 2-cis,4-trans xanthosine, exhibited the highest expression levels at ZY_2, ZY_3 and ZY_4, while LUT5 in the production of zeaxanthin was mainly up-regulated at ZY_2 and ZY_4 (Fig 7A). The downstream enzyme of GA, namely, gibberellin 2-oxidase (GA2ox), played a key role in the metabolic pathway of GA, involving the transformation of GA20 to GA29, GA9 to GA51, GA4 to GA34, and GA1 to GA8, as well as the catabolic reactions of four GA isoforms, GA29, GA51, GA34 and GA8 (Fig 7B). Results indicated that the expression levels of GA2ox was up-regulated at ZY_2, ZY_4 and ZY_5, implying obvious GA synthesis and metabolism at these stages. IAA is formed via tryptophan metabolism, which involves two main reaction pathways. Using tryptophan as a substrate, one pathway is catalyzed by aromatic-L-amino-acid (DDC) and aldehyde dehydrogenase (ALDH), while the other pathway is catalyzed by L-tryptophan-pyruvate aminotransferase (TAA1) and indole-3-pyruvate monooxygenase (YUCCA). The results showed that the common feature of gene expression of these four enzymes (DDC, TAA1, ALDH, and YUCCA) was overexpression at both ZY_2 and ZY_4 stages (Fig 7C). At different growth stages, the JA metabolic pathway in the bulbils mainly involved five enzymes encoded by 16 DEGs, and these enzymes showed differences in heat maps at different stages. The substrate lecithin produced α-linolenic acid under the action of secreted phospholipase A2 (PLA2), and then reacted with a series of enzymes such as lipoxygenase 2S (LOX2S), allene oxide synthase (AOS), 3-hydroxyacyl-CoA dehydrogenase (MFP2) and acetyl-CoA acyltransferase (fadA) to produce JA. The expression levels of DEGs revealed that the expressions of most genes were mainly up-regulated at ZY_2, ZY_3 and ZY_4 (Fig 7D).

## Transcription factors

At different growth stages, TF coding genes of 5 samples were differentially expressed. According to statistics, there were 155 and 293 TF coding DEGs in the two comparison groups, ZY_2 vs. ZY_1 and ZY_5 vs. ZY_4, respectively, all involving 27 families. In addition, there were 223 and 186 TF coding DEGs in the two comparison groups, ZY_3 vs. ZY_2 and ZY_4 vs. ZY_3, respectively, all involving 25 families. AP2/ERF-ERF, bHLH, MYB-related, NAC, WRKY, C3H, C2H2, and MYB families were eight TF families with the largest number of DEGs (Fig 8A). The expressions of most of AP2/ERF-ERF, bHLH, WRKY and MYB coding genes were up-regulated at ZY_3 and ZY_5, which were markedly higher than those at the other three stages. Additionally, the expressions of NAC and C3H coding genes were raised at ZY_4, while other genes were up-regulated at ZY_3 and ZY_5. Moreover, the expressions of C2H2

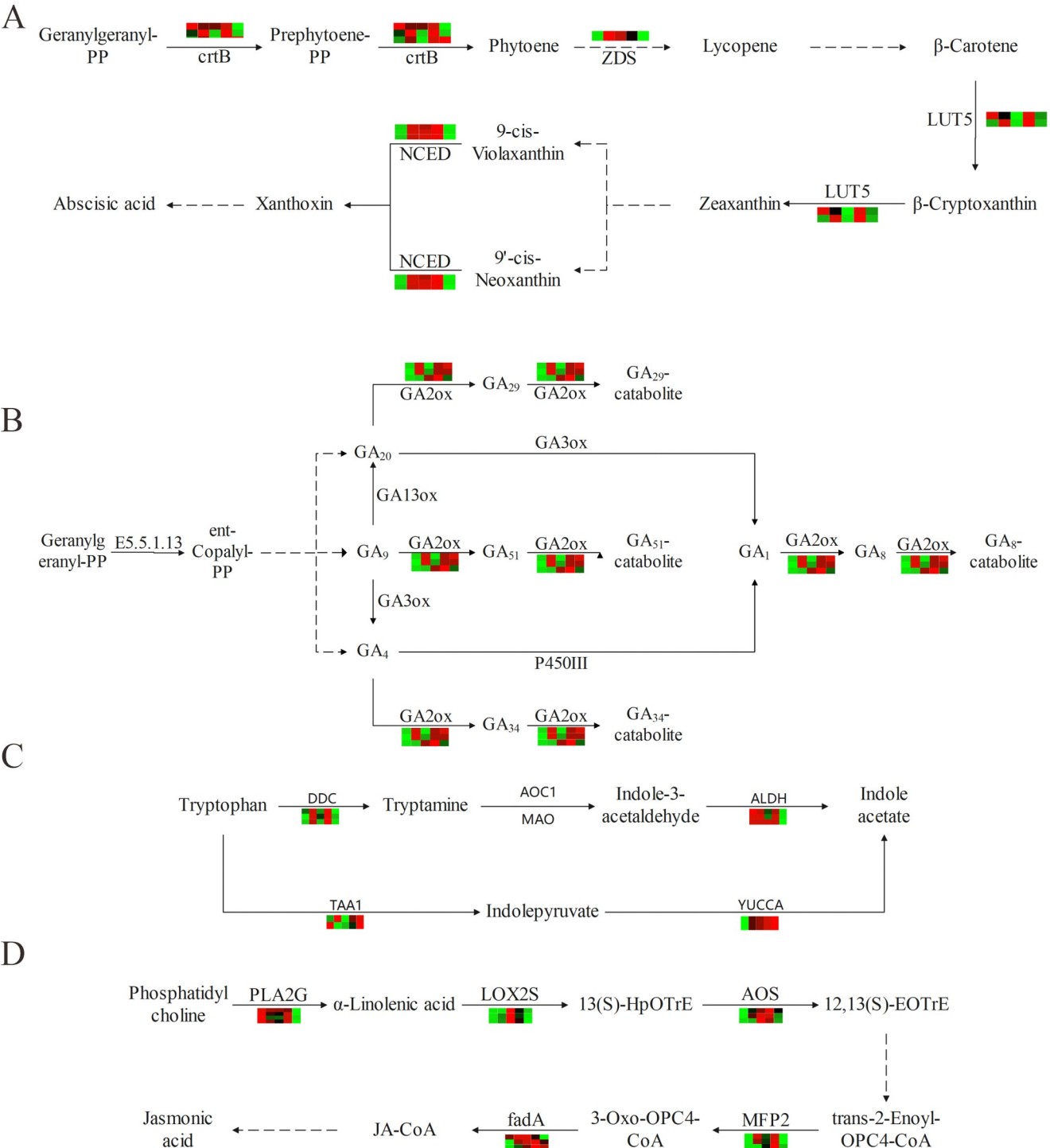

**Fig 7. Heat maps of DEGs related to endogenous hormone metabolism in the bulbils at different growth stages plotted by log2 transformation of FPKM values.** A. ABA. B. GA. C.IAA. D.JA.

coding genes were up-regulated at ZY_2, ZY_3 and ZY_5, and those of MYB-related family coding genes were up-regulated at five stages, but the expressions of most genes were up-regulated mainly at ZY_3 and ZY_5 (Fig 8B).

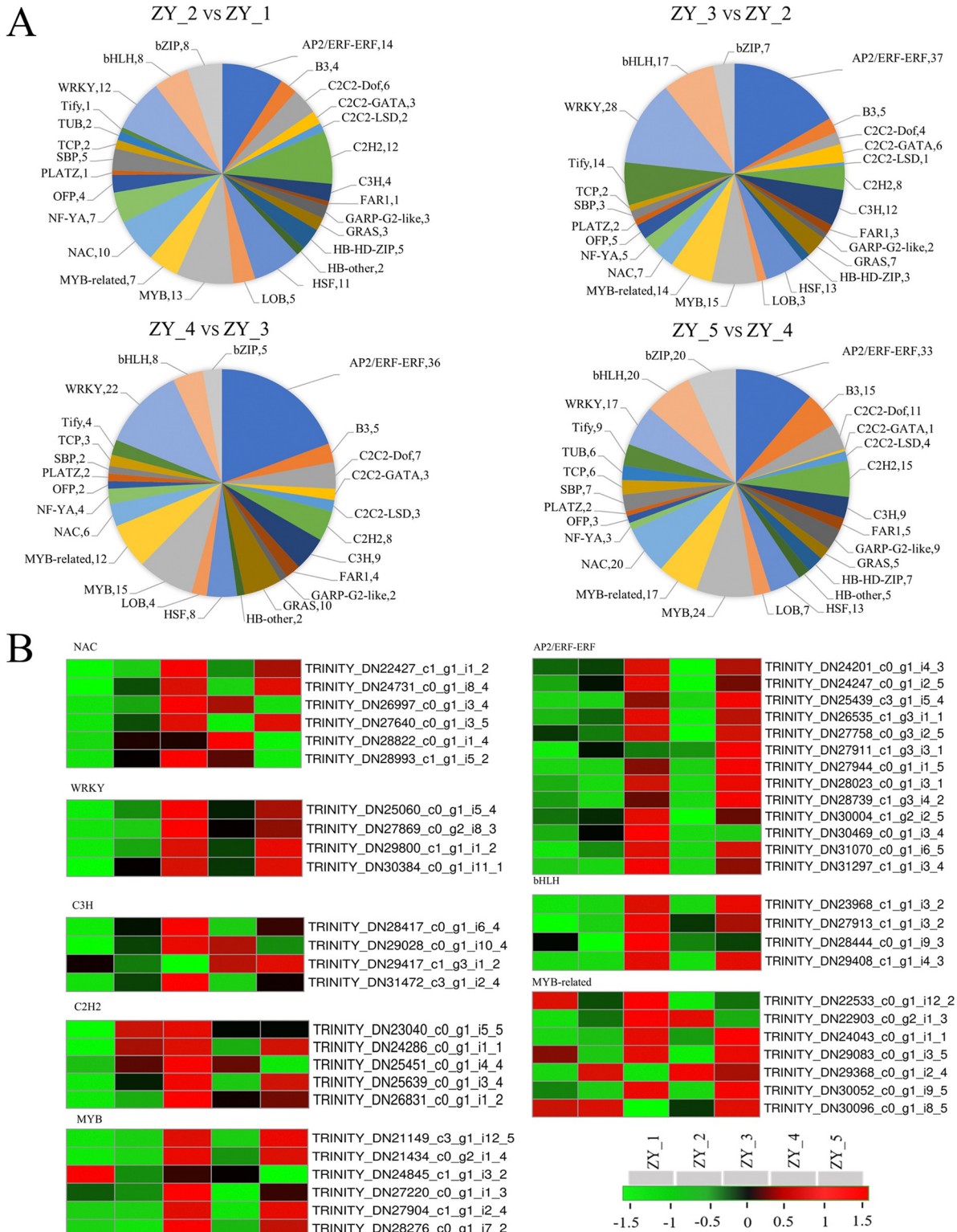

**Fig 8. DEGs related to the TF family in the bulbils at different growth stages.** A. The number of DEGs at different growth stages. B. Heat maps of DEGs at each stage.

## Plant hormone signal transduction

Plant endogenous hormones are able to regulate the growth and development of plants by binding to specific protein receptors, and play a small but efficient regulatory role in plant morphogenesis and biological metabolism. Among them, IAA was directly involved in the regulation process in the nucleus, while the other five hormones played a regulatory role in the nucleus after signal transduction in the cytoplasm (Fig 9A). In this study, the differential expressions of protease genes that can affect DNA expressions were analyzed. In the IAA signal transduction pathway, ARF gene was up-regulated at ZY_1, ZY_2, ZY_3, and ZY_4 and markedly down-regulated at ZY_5. A-ARR, synthesized by transcription and translation, played a physiological role in the nucleus. Its coding genes were mainly raised at ZY_1, ZY_3 and ZY_5, which were notably higher than those at the other two stages. In the process of GA signal transduction, TFs were dissociated by DELLA to inhibit DNA expressions, and most genes were up-regulated at ZY_3 and ZY_5. ABF entered the nucleus from the cytoplasm and acted directly on DNA. Its coding genes were mainly up-regulated at ZY_2 and ZY_4, and the up-regulation of some genes appeared at ZY_3. Furthermore, EIN3 coding genes were highly expressed at ZY_3 and ZY_5, while MYC2 in the JA signal transduction pathway was notably up-regulated at ZY_3 (Fig 9B).

## Weighted gene co-expression network analysis (WGCNA)

**Construction of co-expression network modules.**   The synergistic relationship between the content of endogenous metabolites of *P. ternata* bulbils and the obtained transcript expression values at different growth and development stages were investigated, and the transcripts closely related to endogenous metabolites were speculated. To this end, 15 samples, with a total of 68,386 transcripts, were selected to construct the whole-gene co-expression network. After the genes with low variation in expression (standard deviation ≤0.5) were excluded, the remaining 39,074 genes were finally used to construct the co-expression network. Through hierarchical clustering of dissTOM matrix (Fig 10A), a total of 18 co-expression clustering modules were obtained (Fig 10B). Among these modules, the grey module (Grey) was a set of genes that could not be assigned to any module and showed no reference significance. The blue2 module (Fig 10C) and the dark olive green module (Fig 10D) were selected to display the transcript expression profiles constructed by the co-expression network, respectively, under the co-expression clustering modules. The major co-expressed genes in each module are shown in S2 Table.

**Screening of co-expressed genes of endogenous metabolites in the bulbils at different growth stages.**   Among the 18 co-expression clustering modules obtained, antique white 2、blue 2、dark olive green and honey dew modules displayed significantly positive correlations with the content of endogenous metabolites in the bulbils (Fig 11A). Close correlations of the antique white 2 module with the content of IAA (R = 0.92, $p<0.001$) and ABA (R = 0.92, $p<0.001$) were observed. The blue2 module, showed extremely marked positive correlations with JA (R = 0.77, $p<0.001$) and ABA (R = 0.79, $p<0.001$). The dark olive green module was positively correlated with starch (R = 0.86, $p<0.001$), and the honeydew module displayed the closest correlation with sucrose (R = 0.96, $p<0.001$)、SPS (R = 0.94, $p<0.001$)、SuSy (R = 0.84, $p<0.001$)、SS (R = 0.90, $p<0.001$)、JA (R = 0.84, $p<0.001$)、ABA (R = 0.82, $p<0.001$).

Finally, Cytoscape was employed to visualize the module network for the screening of the core genes of each module. It was found that the core gene was TRINITY_DN18564_c0_g1_i2_5 in the antique white 2 module, TRINITY_DN13478_c0_g1_i1_3 in the blue 2 module, TRINITY_DN27378_c0_g1_i3_2 in the dark olive green module, and

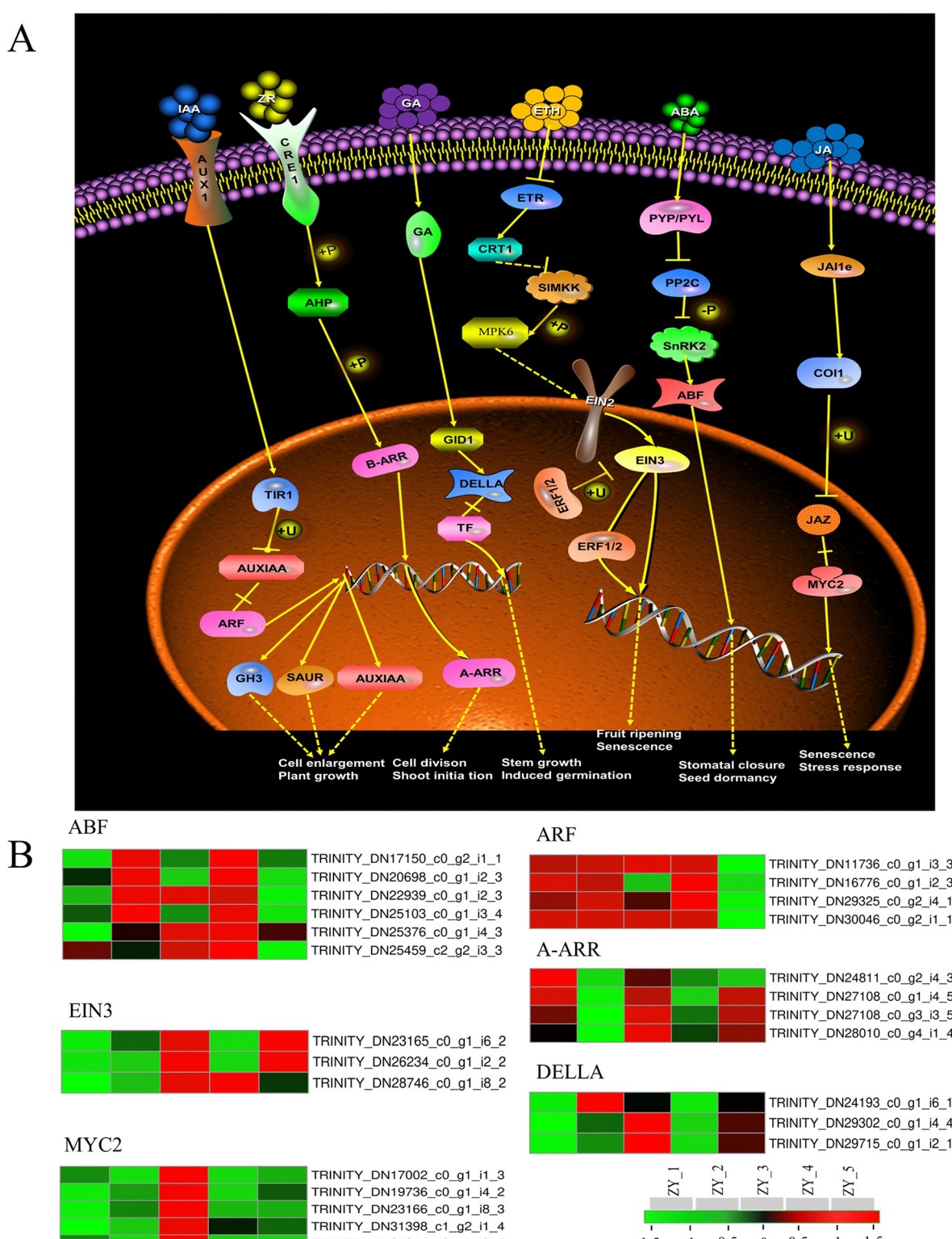

**Fig 9. DEGs related to plant hormone signal transduction in the bulbils at different growth stages.** A. Plant hormone signal transduction pathway B. Heat maps of DEGs at different growth stages.

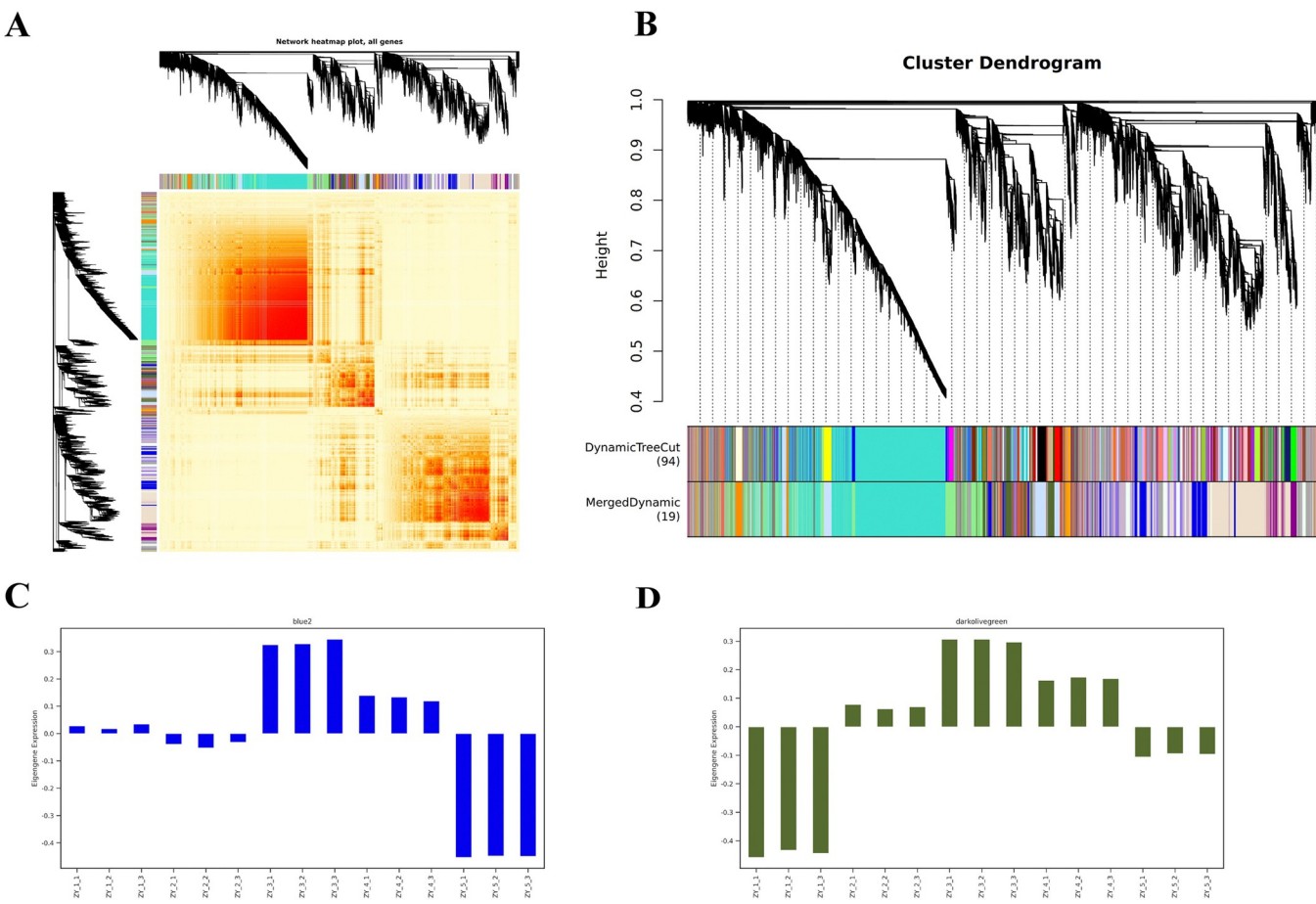

**Fig 10. Gene co-expression network analysis of 15 samples of bulbils at different growth stages.** A. TOM correlation heat maps for all transcripts. B. Co-expression network modules constructed by transcripts. Different colors represent the network modules constructed, and the phylogenetic tree represents the hierarchical clustering of different samples, where each color corresponds to one module. C. Transcript expression profiles in the co-expressed blue2 module. D. Transcript expression profiles in the co-expressed dark olive green module. In the histogram, the x-axis represents all samples, and the y-axis represents the expression profile of the transcripts in the module.

TRINITY_DN24626_c0_g1_i2_3 in the honeydew module (Fig 11B-11E). These genes are annotated as hypothetical proteins in the NR database. Further comparison of their RNA sequences with NCBI Reference Sequence indicates that the structure and function of the proteins encoded by these genes have not been completely resolved. These results suggest that the analysis of the structure and function of unknown proteins may be a key way to further understand the regulatory network of genes related to the characteristics of bulbils development.

## Discussion

Bulbils of *P. ternata* is an important raw material for medicinal use. At present, the understanding of the growth and development characteristics of *P. ternata* bulbils and the gene regulation network is incomprehensive. It is of great importance to deeply understand the growth and development and molecular mechanisms of bulbils for protecting and developing its medicinal resources. In this study, the results of the above experiments were synthesized based on physiological and biochemical responses and biological information.

Bulbils are vital storage organs and reproductive organs of *P. ternata*. Sucrose and starch are the end products of photosynthesis, which are responsible for the transportation and

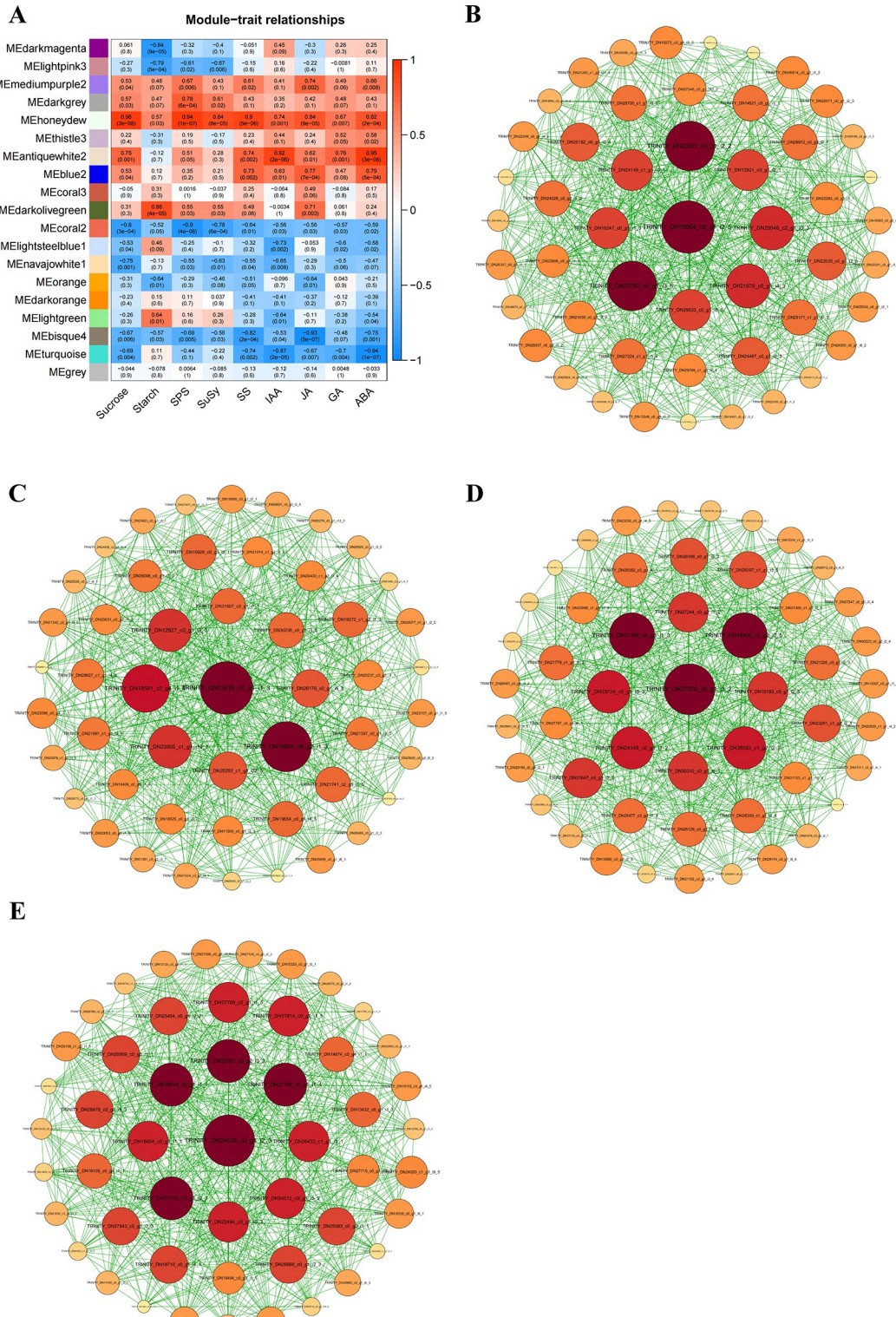

**Fig 11. Core transcripts that were significantly associated with changes in endogenous metabolite content of sucrose, starch, SPS, SuSy, SS, IAA, JA, GA and ABA were screened in the co-expression module.** A. Correlation heat maps of the identified co-expression module with the content of endogenous metabolites in the bulbils. B. Gene co-expression network of the identified antiquewhite2 module related to the content of IAA and ABA. C. Gene co-expression network of the identified blue2 module related to the content JA and ABA. D. Gene co-expression network of the identified dark olive

green module related to the content of starch. E. Gene co-expression network of the identified honeydew module related to the content of sucrose, JA, ABA and SPS, SuSy, SS.

storage of carbohydrates in higher plants, respectively, and they are also signaling molecules that coordinate the relationship between source and sink of plants [39, 40]. More and more evidence shows that sucrose is able to regulate intracellular metabolism at the level of gene expression [41–43], and wide attention has been paid to its vital role in plant growth and development and assimilate distribution schemes [44]. In the processes of growth and development, plants often adapt to the environment through the spatiotemporal gene expression, and show corresponding botanical characteristics. SPS is a key rate-limiting enzyme during sucrose synthesis [45, 46], and SS is a key enzyme that decomposes sucrose and serves as a substrate for starch synthesis [47]. SPS is involved in the carbon allocation between sucrose and starch [48], and with high activity, it increases the sucrose content and decreases starch synthesis [49]. In this study, SPS activity was significantly increased at both ZY_2 and ZY_4 stages. Sucrose and starch contents were also significantly increased in both ZY_2 and ZY_4. The changes in the activities of SuSy and SS were characterized similarly to the changes in the contents of sucrose and starch, respectively. Among the metabolic pathways of sucrose and starch of concern, the carbon required by bulbils was mainly obtained by the transportation of the mother stem at the early stage and the photosynthesis of leaves after emergence. At the first four stages, INV coding genes were continuously highly expressed. They transformed the carbon source transported from the mother stem to the bulbils and the assimilates produced by photosynthesis into D-glucose 6-phosphate and fructose, which were finally synthesized into starch under the action of downstream genes. Among them, the coding genes of eight enzymes, namely, E2.4.1.14, SUS, UGP2, pgm, glgC, glgA, WAXY and GBE1, were evidently up-regulated at ZY_2 and ZY_4 in terms of expression, which was identical to the variation trend of the starch content. This result indicated that ZY_2 and ZY_4 are crucial stages of starch accumulation during the growth and development of bulbils.

Plant hormones are chemical messengers, which can regulate the growth and development of plants and increase their biomasses [50, 51]. As plant hormones, ETH and JA exhibit defensive functions, which can respond to external stimuli and improve the immunity of plants [52]. In this study, the content of jasmonic acid generally increased from ZY_1 to ZY_4 stage, and was significantly lower in ZY_5 stage than in other stages, suggesting that jasmonic acid enhanced the adaptability of bulbils to the environment during the transition from tender tissue to mature tissue. These results support the results of related enzyme gene expression levels. Among GAs, $GA_1$, $GA_3$, $GA_4$ and $GA_7$ have been proven to have a wide range of biological activities [53]. GA2ox was differentially expressed in the GA metabolic pathway at different stages, involving $GA_1$, $GA_4$, $GA_9$, $GA_{20}$, $GA_{29}$, $GA_{34}$ and $GA_{51}$. There was no significant difference in gibberellin content between ZY_1 and ZY_4 stages, and the content of gibberellin remained at a low level, indicating that the low concentration of GA in the natural state was conducive to the normal growth of the bulbils. IAA regulates the growth and development of plants and the stress response to external environmental stimuli, and it often plays a coordinated role with other hormones [54]. In this study, the content of IAA in bulbils decreased during the whole growth period, and auxin content was highest only at the initial formation of bulbils (ZY_1 stage) and when new bulbils was formed from the stalk of germination of bulbils (ZY_4 stage), while the content was lower in other periods. Therefore, it is speculated that IAA may have a strong regulation effect during the formation of bulbils. The regulation was weakened in the growth stage of bulbils, which was in good agreement with the expression levels of related genes in the IAA metabolic pathway. ABA is a type of growth-inhibiting hormone

involved in regulating various physiological processes [55]. ABA content showed an increasing trend before the ZY_4 stage, and significantly increased in the ZY_4 stage. Along with the whole expansion and growth stage of the bulbils, the variation characteristics of ABA content might be related to the regulation of the distribution and accumulation of carbohydrates in *P. ternata.*

TFs play a crucial role in plant morphogenesis, as they act as a magical key that unlocks the multiple codes of plant development and growth. The AP2/ERF family, as one of the largest TF families in plants, is widely involved in plant growth and development, biosynthesis and various stress responses. AP2/ERF-ERF is one of its subfamilies (ETH response factor). Many ERF subfamilies have been proven to promote plant growth and development, and improve tolerance to hormone and abiotic stress in maize, Arabidopsis thaliana, wheat and tobacco [56]. In addition, some other large TF families were also found in our research, such as bHLH, MYB-related, NAC, WRKY, C3H, C2H2 and MYB families. The coding genes of these TFs were differentially expressed in the heat maps at different stages, especially at ZY_3 and ZY_5, during which their expressions were evidently up-regulated. Interestingly, a common feature is shared by these TFs, i.e., they are large families comprising a large number of members, which can participate in plant growth and development and biological metabolism, and respond to external environmental stress. For example, TFs in the bHLH family are generally involved in plant growth and metabolism, photomorphogenesis, light signal transduction and secondary metabolism, and can improve the resistance of plants to environmental stress [57, 58]. In the MYB-related family, LlMYB3 protein is overexpressed in transgenic Arabidopsis thaliana, which confirms ABA hypersensitivity and indicates the enhanced tolerance to cold, drought and salt stresses. LlMYB3 may also be involved in the anthocyanin synthesis pathway [59]. Some uncommon TFs, such as C3H and C2H2, are essential in plant stress responses [60, 61], although the regulatory network is still unclear. The number and function of TFs are often quite different in different plants, and there is little research on TFs in P. ternata. The results of this study manifested that ZY_3 and ZY_5 were two distinct growth and development stages of the bulbils, when many TFs were up-regulated intensively.

Phytohormones play a key role not only in abiotic tolerance but also are deeply involved in the regulation of plant morphological and metabolic responses by mediating a wide range of responses. The transduction of the IAA signal depends on three protein families: Aux/IAA, TIR1, and ARF. TIR1 negatively feedback regulates IAA signaling by inhibiting Aux/IAA, and high concentrations of IAA accelerate protein hydrolysis and alleviate inhibition of growth hormone-responsive genes by promoting the formation of ARF dimers. Thus high levels of IAA can release ARF activity. In this study, the coding genes of ARF was highly expressed in ZY_1, ZY_2 and ZY_4 stages. In the signaling pathway of zeatin riboside (ZR), the genes encoding the A-ARR protein is highly expressed in the ZY_1, ZY_3, and ZY_5 stages. DELLA protein is inhibitors of GA signaling and regulate the GA signaling process through negative feedback. High concentrations of GA increase the activity of DELLA protein. ABF is a critical positive regulator of ABA signaling. In this study, ABA content was increased and ABF expression was elevated. The two were positively correlated. In the JA signaling pathway, JAZ repressors negatively regulate jasmonate signaling by suppressing transcription factors. MYC2 is a bHLH transcription factor that targets jasmonate-responsive genes and regulates jasmonate-mediated processes [62].

At last, based on WGCNA, the complete transcriptome co-expression network for phenotypic traits was constructed, and four core transcripts involved in the bulbils development process were screened.

## Conclusions

This study found that the ZY_2 and ZY_4 stages are the critical periods for bulbils to accumulate sucrose and starch, and the changes in sucrose and starch contents are basically consistent with the activities of related enzymes and the differential expression characteristics of enzyme genes in their metabolic pathways. Endogenous hormones are essential for the development of bulbils, and the differential expression characteristics of key enzyme genes in their metabolic pathways well support the measurement results of their contents. Eight TF families were significantly up-regulated in the ZY_3 and ZY_5 stages. Four core transcripts involved in the development process of bulbils have been screened by WGCNA. The structures and functions of proteins encoded by these transcripts have not been fully analyzed yet, indicating that analyzing the structures and functions of these unknown proteins may be a key approach to gain a deeper understanding of the gene regulatory network related to bulbils development. Finally, the reliability of transcriptome results was verified by qRT-PCR analysis. This study provides an information source for analyzing the molecular mechanism of bulbils growth and development, and also helps to solve the problem of lack of genetic information in non-model plant species.

## Supporting information

**S1 Table. Gene primers used in the reverse transcription-quantitative PCR (RT-qPCR) analysis.**
(XLSX)

**S2 Table. Detailed gene annotation.**
(XLSX)

**S3 Table. Functional annotation of overexpressed genes.**
(XLSX)

**S4 Table. Antiquewhite2_major co-expressed genes.**
(XLSX)

**S5 Table. Blue2_major co-expressed genes.**
(XLSX)

**S6 Table. Darkolivegreen_major co-expressed genes.**
(XLSX)

**S7 Table. Honeydew_major co-expressed genes.**
(XLSX)

**S1 Fig. Plant indole-3-acetic acid (IAA).**
(PDF)

**S2 Fig. Plant abscisic acid (ABA).**
(PDF)

**S3 Fig. Plant gibberellin (GA).**
(PDF)

**S4 Fig. Plant jasmonic acid (JA).**
(PDF)

**S5 Fig. Plant sucrose phosphate synthase (SPS).**
(PDF)

**S6 Fig. Plant sucrose synthase.**
(PDF)

**S1 Text. Sequences of overexpressed genes.**
(DOCX)

**S2 Text. Gene sequence.**
(FA)

## Acknowledgments

Thank Gansu University of Chinese Medicine for providing the research experiment platform, and also thank all the authors for their contributions to this research.

## Author Contributions

**Conceptualization:** Xiwei Jia, Honggang Chen, Tao Du.

**Data curation:** Xiwei Jia, Xijia Jiu, Yuan Liu.

**Formal analysis:** Xiwei Jia, Xijia Jiu.

**Funding acquisition:** Honggang Chen, Tao Du.

**Investigation:** Xiwei Jia, Xijia Jiu, Yuan Liu, Chao Guo, Dong Liu, Honggang Chen.

**Methodology:** Xijia Jiu, Dong Liu, Honggang Chen, Tao Du.

**Project administration:** Honggang Chen, Tao Du.

**Resources:** Xijia Jiu, Yuan Liu, Chao Guo, Dong Liu, Xin Zhao, Tao Du.

**Software:** Xiwei Jia.

**Supervision:** Honggang Chen, Tao Du.

**Validation:** Honggang Chen, Tao Du.

**Visualization:** Honggang Chen, Tao Du.

**Writing – original draft:** Xiwei Jia, Xijia Jiu, Tao Du.

**Writing – review & editing:** Xijia Jiu, Honggang Chen, Tao Du.

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
