## [Decision Letter · Decision Letter 0]

13 Aug 2024

PONE-D-24-28450Analysis of key gene networks controlling the characteristics of Pinellia ternata Bulbils development by transcriptome and physiological and biochemical responsesPLOS ONE

Dear Dr. Du,

Thank you for submitting your manuscript to PLOS ONE. After careful consideration, we feel that it has merit but does not fully meet PLOS ONE’s publication criteria as it currently stands. Therefore, we invite you to submit a revised version of the manuscript that addresses the points raised during the review process.

We look forward to receiving your revised manuscript.

Kind regards,

Arun Kumar Shanker

Academic Editor

PLOS ONE

Journal Requirements:

This work was supported by the Science and Technology Key R&D Program in Gansu Province (21YF5NA130) and the Special Foundation for Construction of National Traditional Chinese Medicine Industry Technology System in China "Supported by the earmarked fund for CARS-21".

4. Thank you for uploading your study's underlying data set. Unfortunately, the repository you have noted in your Data Availability statement does not qualify as an acceptable data repository according to PLOS's standards.

**Additional Editor Comments:**

We have now received and reviewed the reports from two of our reviewers. One of the reviewers has raised pertinent concerns regarding the manuscript, which we believe are crucial to address in order to improve the quality and clarity of your work.

We kindly request that you revise your manuscript (MS) in accordance with the suggestions provided by the reviewers. It is important that you prepare a detailed point-by-point response to each of the comments, indicating how you have addressed them in your revised manuscript. If there are any comments with which you disagree, please provide a clear and reasoned explanation in your response.

Reviewers' comments:

Reviewer's Responses to Questions

**Comments to the Author**

1. Is the manuscript technically sound, and do the data support the conclusions?

Reviewer #1: Yes

Reviewer #2: Partly

Reviewer #3: Yes

2. Has the statistical analysis been performed appropriately and rigorously? 

Reviewer #1: Yes

Reviewer #2: No

Reviewer #3: Yes

3. Have the authors made all data underlying the findings in their manuscript fully available?

Reviewer #1: Yes

Reviewer #2: Yes

Reviewer #3: Yes

4. Is the manuscript presented in an intelligible fashion and written in standard English?

Reviewer #1: Yes

Reviewer #2: No

Reviewer #3: No

5. Review Comments to the Author

**Reviewer #1:** The detailed comments are mentioned in the reviewed manuscript file

1. Write the full scientific name wherever the item is mentioned first time

2. Please provide mechanisms where mentioned in comments

3. Please provide detailed genetic annotations in supplimentry files with sequences

4. Please improve english language. And remove arbitrary words such as generally speaking, fisrtly, secondly etc

**Reviewer #2:** The manuscript is written surprisingly without any citation except in some very few sentences.

i) At first it should rejected because no supporting documents were there.

ii) It does not have representation of proper statistical tables of results by which data can be comprehended.

iii) Why this research is important is not legibly stated, they repeated the experiment stated details of bulbil development is done through transcriptome anaylis under dfferent stages. What are the enzymes or proteins that have medicinal properties of this plant?

iv) If any citation of discovery of any alkaloid or protein or enzyme is there it should be reflected in the manuscript. It is not found.

v) Is it very important to know the bubil development (Significance of research/ objective of research/ outcome of research) until any targetted protein which has some unique medicinal property is found and its mode of action can be unfolded?

vi) Activity starch or other enzymes, activity of different enzymes like Auxin, Gibberelin, Cytokinin in different stages are found in their study and with mode of action of diffrent genes by transcriptome analysis. But the significance of research is not found in the manuscript.

As a whole I would reject the manuscript for publishing in this journal.

**Reviewer #3:** Comment 1:

The title is clear and accurately reflects the content of the study. However, consider revising the title for conciseness. For example, "Transcriptomic and Biochemical Insights into Key Gene Networks Driving Bulbil Development of Pinellia ternata " might be more succinct.

Comment 2:

Line No: 16 - The phrase "is is" is repeated in the first sentence.

Comment 3:

The abstract generally conveys the study's objectives and findings well, but some sentences could be clearer. For example, "It is of great significance to deeply understand the growth and development laws and molecular mechanisms of bulbils" could be simplified to "Understanding the growth, development, and molecular mechanisms of bulbils is crucial."

Comment 4:

The abstract is heavy on technical terms (e.g., WGCNA, SPS, SuSy, SS, ABA, JA, IAA) without explanations. While these terms may be familiar to specialists, a brief explanation or context for less common terms would make the abstract more accessible.

Comment 5:

The abstract transitions between topics somewhat abruptly. For instance, after discussing the biochemical findings, it quickly moves to RNA-Seq analysis without explaining the connection between the biochemical changes and the transcriptomic analysis. Adding a sentence that links these sections would improve flow.

Comment 6:

The abstract requires proper conclusion and abstract section is slightly lengthy and could benefit from conciseness. Reducing redundancy and simplifying complex sentences would make it more reader friendly.

Comment 7:

The introduction provides a broad overview of the medicinal value and botanical characteristics of P. ternata. However, it could be more focused by clearly outlining the specific research problem earlier on. For instance, the introduction could start with a brief discussion on the importance of understanding bulbil development in P. ternata and then delve into its medicinal uses.

Comment 8:

The introduction covers a wide range of background information, from the medicinal properties of P. ternata to the role of bulbils in asexual reproduction. While informative, the text could benefit from better organization. Consider grouping related information together, such as discussing the medicinal uses and chemical constituents of P. ternata in one section and the role of bulbils and their biochemical pathways in another.

Comment 9: There are a few grammatical errors and stylistic issues that could be addressed. For instance, the phrase "bead buds" is used interchangeably with "bulbils," which could confuse readers. Consistency in terminology would improve readability. Also improve the flow of introduction with objectives of the study and strong concluding statement for the introduction.

Comment 10:

A critical detail missing from the "Materials and Methods" section is the identification of the specific variety, landrace, or type of Pinellia ternata used in the study.

Comment 11:

The term “Irregular observations” in line no: 133 and 134 could be better clarified—does it mean observations were made as needed, or were there specific criteria?

Comment 12:

The use of enzyme-linked immunosorbent assay (ELISA) to determine hormone levels and enzyme activities is appropriate and common in such studies. The specific kits used are mentioned, which aids in replicability. However, the conditions for centrifugation (1000×g for 20 minutes) seem a bit low for typical sample processing, where higher speeds are often used. The rationale behind this choice might be worth exploring.

Comment 13:

The description of RNA isolation, DNA digestion, and cDNA synthesis is generally sufficient. However, it would be better if the kit or method used for RNA extraction was specified.

Comment 14:

In Bioinformatics Analysis: The use of Trimmomatic for data trimming and Trinity for de novo assembly is well-established. However, the versions of the software and specific parameters used (e.g., minimum quality score, length of reads retained) should be provided.

Comment 15:

The alignment of unigenes with NR, Swiss-Prot, and other databases is a standard approach. The software used for expression level calculation (bowtie2 and eXpress) is mentioned, but again, specific parameters and versions would be beneficial.

Comment 16:

In Validation Process: The use of qRT-PCR to validate DEGs is a solid approach. The two-step process is clearly described, though it might be helpful to include information on the primer sequences used for each gene and the reference genes for normalization.

Comment 17:

The statistical analysis is well-detailed, with ANOVA and LSD tests applied. Mentioning that there were three biological replicates adds robustness to the analysis. However, it's important to ensure that the data meet the assumptions of ANOVA (e.g., normality, homogeneity of variance).

Comment 18: Another important aspect to address in the "Materials and Methods" section is the detailed explanation of the software used for data analysis and graph creation. It's important to specify the software versions, any custom scripts or parameters applied during analysis, and the exact methods used to generate the graphs (e.g., which software or programming language was used, any specific libraries or packages, and how the data was visualized). This level of detail ensures that other researchers can replicate the analysis and produce similar visualizations, enhancing the transparency and credibility of the study.

Comment 19: It would be beneficial to rearrange the "Results and Discussion" section to follow a logical sequence that aligns with the flow of the study. This restructuring will help readers follow the progression of the research more easily and understand the connections between different findings. Additionally, attention should be given to correcting any grammatical errors throughout this section to ensure clarity and professionalism in the presentation of the results.

6. PLOS authors have the option to publish the peer review history of their article (what does this mean?). If published, this will include your full peer review and any attached files.

Reviewer #1: **Yes: **Shamshir ul Hussan

Reviewer #2: No

Reviewer #3: **Yes: **Vijayakumar Eswaramoorthy

---

## [Author Response · Author response to Decision Letter 0]

17 Sep 2024

Response to Reviewers

We thank the editors and reviewers for their guidance and helpful comments. According to suggestions, we have made a lot of changes to the full text, Includes language editing, reopening the discussion, modifying inappropriate descriptions, rethinking and summarizing results, and point-by-point responses to each comment.

Reviewer #1: 

1. Write the full scientific name wherever the item is mentioned first time.

Authors’ Response: The "item" first mentioned in the paper has been given a full scientific name based on the reviewer's suggestion.

2. Please provide mechanisms where mentioned in comments.

Authors’ Response: The mechanisms mentioned in the comments have been supplemented in the material and method section.

3. Please provide detailed genetic annotations in supplimentry files with sequences.

Authors’ Response: The genetic annotations of the sequences have been supplemented in supplimentry files (Table S2).

4. Please improve english language. And remove arbitrary words such as generally speaking, fisrtly, secondly etc.

Authors’ Response: The grammar and diction problems raised by reviewers have been improved in the paper.

Reviewer #2: 

1. At first it should rejected because no supporting documents were there.

Authors’ Response: Thanks for the reviewer's comments, but I'm sorry that I can't understand the supporting documents stated in the comments.

2. It does not have representation of proper statistical tables of results by which data can be comprehended.

Authors’ Response: Thanks to the reviewer for comments. The data results obtained through the experiment are reflected in the form of graphs in the paper, following the principle of concise and clear. In addition, sample collection method, collection time, and biological duplication are clearly described in the paper, but I do not understand what the statistical table of results stated in the reviewer's comments refers to specifically.

3. Why this research is important is not legibly stated, they repeated the experiment stated details of bulbil development is done through transcriptome anaylis under dfferent stages. What are the enzymes or proteins that have medicinal properties of this plant？

4. If any citation of discovery of any alkaloid or protein or enzyme is there it should be reflected in the manuscript. It is not found.

Authors’ Response: The chemical composition of Pinellia ternata bulbils is very complex, and a series of current reports on it have elaborated its true medicinal efficacy. However, medicinal efficacy research and pharmacological research are not my research areas. I only summarized the existing medicinal value in the introduction to illustrate its important medicinal value and potential. In my research area, I am more concerned with the growth process and developmental characteristics of bulbils.

5. Is it very important to know the bubil development (Significance of research/ objective of research/ outcome of research) until any targetted protein which has some unique medicinal property is found and its mode of action can be unfolded？

Authors’ Response: There are many research reports on the medicinal value of Pinellia ternata, but how it exerts its medicinal effect and through what mode it exerts its efficacy are not my concerns. However, as a medicinal plant and a raw material for natural medicine, it is necessary to pay attention to its growth process and developmental characteristics.

6. Activity starch or other enzymes, activity of different enzymes like Auxin, Gibberelin, Cytokinin in different stages are found in their study and with mode of action of diffrent genes by transcriptome analysis. But the significance of research is not found in the manuscript.

Authors’ Response: Through the above data analysis, the purpose is to explore the growth and development characteristics of Pinellia ternata bulbils, a botanical medicine, and understand its gene expression pattern, which can provide information sources for the analysis of the molecular mechanism of the growth and development of bulbils, and also help solve the problem of the lack of genetic information of non-model plant species.

Reviewer #3: 

1. The title is clear and accurately reflects the content of the study. However, consider revising the title for conciseness. For example, "Transcriptomic and Biochemical Insights into Key Gene Networks Driving Bulbil Development of Pinellia ternata " might be more succinct.

Authors’ Response: Many thanks to the reviewer for the valuable suggestion and the title of the paper has been revised in “Revised Manuscript with Track Changes”.

2. Line No: 16 - The phrase "is is" is repeated in the first sentence.

Authors’ Response: Changes have been made in the original text in response to the reviewers' suggestions.

3. The abstract generally conveys the study's objectives and findings well, but some sentences could be clearer. For example, "It is of great significance to deeply understand the growth and development laws and molecular mechanisms of bulbils" could be simplified to "Understanding the growth, development, and molecular mechanisms of bulbils is crucial."

Authors’ Response: Thank you for your comment! Changes have been made in the original text in response to the reviewers' suggestion.

4. The abstract is heavy on technical terms (e.g., WGCNA, SPS, SuSy, SS, ABA, JA, IAA) without explanations. While these terms may be familiar to specialists, a brief explanation or context for less common terms would make the abstract more accessible.

Authors’ Response: Thank you for your honest advice on this issue, Abbreviations for specialized terms in the abstract section are given their full scientific names in the introduction section and later in the text when they first appear.

5. The abstract transitions between topics somewhat abruptly. For instance, after discussing the biochemical findings, it quickly moves to RNA-Seq analysis without explaining the connection between the biochemical changes and the transcriptomic analysis. Adding a sentence that links these sections would improve flow.

Authors’ Response: In response to the reviewer's suggestion, we have revised the “Results” section of the paper to meet the reviewer's requirements.

6. The abstract requires proper conclusion and abstract section is slightly lengthy and could benefit from conciseness. Reducing redundancy and simplifying complex sentences would make it more reader friendly.

Authors’ Response: The “Abstract” section of the paper has been supplemented and improved at the request of the reviewers.

7. The introduction provides a broad overview of the medicinal value and botanical characteristics of P. ternata. However, it could be more focused by clearly outlining the specific research problem earlier on. For instance, the introduction could start with a brief discussion on the importance of understanding bulbil development in P. ternata and then delve into its medicinal uses.

8. The introduction covers a wide range of background information, from the medicinal properties of P. ternata to the role of bulbils in asexual reproduction. While informative, the text could benefit from better organization. Consider grouping related information together, such as discussing the medicinal uses and chemical constituents of P. ternata in one section and the role of bulbils and their biochemical pathways in another.

Authors’ Response: Thank you very much for the comment. In response to the reviewer's comment, we have Revised and improved the writing logic and order problems in the "introduction" part of the paper in the "Revised Manuscript with Track Changes" file and highlighted them.

9. There are a few grammatical errors and stylistic issues that could be addressed. For instance, the phrase "bead buds" is used interchangeably with "bulbils," which could confuse readers. Consistency in terminology would improve readability. Also improve the flow of introduction with objectives of the study and strong concluding statement for the introduction.

Authors’ Response: Thanks to this crucial comment, we have standardized the terminology in the paper, e.g., describing “beard buds” as “bulbils”.

10. A critical detail missing from the "Materials and Methods" section is the identification of the specific variety, landrace, or type of Pinellia ternata used in the study.

Authors’ Response: The “Materials and Methods” section of the paper has been supplemented and highlighted as requested by the comment, and this work is reflected in the “Revised Manuscript with Track Changes” document.

11. The term “Irregular observations” in line no: 133 and 134 could be better clarified—does it mean observations were made as needed, or were there specific criteria?

Authors’ Response: The growth and development process of Pinellia ternata Bulbils varies under different climatic conditions in different regions (i.e., the time to reach each stage of growth is different), and the term “Irregular observations” in the paper means that frequent observations were made to determine whether the morphological characteristics of the bulbils met the specific requirements and thus to determine the specific time to take samples. Therefore, the purpose of “Irregular observations” is to accurately obtain samples of bulbils that meet the requirements.

12. The use of enzyme-linked immunosorbent assay (ELISA) to determine hormone levels and enzyme activities is appropriate and common in such studies. The specific kits used are mentioned, which aids in replicability. However, the conditions for centrifugation (1000×g for 20 minutes) seem a bit low for typical sample processing, where higher speeds are often used. The rationale behind this choice might be worth exploring.

Authors’ Response: Thank you for your comments on this issue, the centrifugation conditions were set in full accordance with the kit instructions, this condition has also been used in many studies and some of the references have been cited in the paper, as well as the instructions for the kits used in this study are provided in the Supplementary file.

13. The description of RNA isolation, DNA digestion, and cDNA synthesis is generally sufficient. However, it would be better if the kit or method used for RNA extraction was specified.

Authors’ Response: Additional information on the kits used for RNA extraction and DNA purification is provided in the section “Ribonucleic acid (RNA) isolation, complementary deoxyribonucleic acid (cDNA) database preparation, transcriptome sequencing”.

14. In Bioinformatics Analysis: The use of Trimmomatic for data trimming and Trinity for de novo assembly is well-established. However, the versions of the software and specific parameters used (e.g., minimum quality score, length of reads retained) should be provided.

Authors’ Response: The version of the software and the specific parameters used have been supplemented and highlighted in the “Bioinformatic analysis” section.

15. The alignment of unigenes with NR, Swiss-Prot, and other databases is a standard approach. The software used for expression level calculation (bowtie2 and eXpress) is mentioned, but again, specific parameters and versions would be beneficial.

Authors’ Response: Thanks to the reviewer for this important comment, the software version and specific parameters were added and highlighted in the “Bioinformatic analysis” section.

16. In Validation Process: The use of qRT-PCR to validate DEGs is a solid approach. The two-step process is clearly described, though it might be helpful to include information on the primer sequences used for each gene and the reference genes for normalization.

Authors’ Response: In response to this important comment, information on primer sequences and reference genes has been provided in the Supplementary file (Table S1).

17. The statistical analysis is well-detailed, with ANOVA and LSD tests applied. Mentioning that there were three biological replicates adds robustness to the analysis. However, it's important to ensure that the data meet the assumptions of ANOVA (e.g., normality, homogeneity of variance).

Authors’ Response: Thank you very much for your suggestion, normal distribution is a prerequisite for ANOVA, so we tested all data for normal distribution and all data met the statistical requirements. The test of homogeneity of variance and other complementary tests ensured the reliability of the analyzed results.

18. Another important aspect to address in the "Materials and Methods" section is the detailed explanation of the software used for data analysis and graph creation. It's important to specify the software versions, any custom scripts or parameters applied during analysis, and the exact methods used to generate the graphs (e.g., which software or programming language was used, any specific libraries or packages, and how the data was visualized). This level of detail ensures that other researchers can replicate the analysis and produce similar visualizations, enhancing the transparency and credibility of the study.

Authors’ Response: In response to this important comment, we have added and refined the relevant software versions and usage parameters in the “Materials and Methods” section.

19. It would be beneficial to rearrange the "Results and Discussion" section to follow a logical sequence that aligns with the flow of the study. This restructuring will help readers follow the progression of the research more easily and understand the connections between different findings. Additionally, attention should be given to correcting any grammatical errors throughout this section to ensure clarity and professionalism in the presentation of the results.

Authors’ Response: We would like to thank the reviewer for the crucial comments, and in response to these suggestions, we have reorganized the Discussion section in accordance with the Analysis of Results section and have checked and corrected grammatical errors throughout the text.

---

## [Decision Letter · Decision Letter 1]

11 Oct 2024

PONE-D-24-28450R1Transcriptomic and Biochemical Insights into Key Gene Networks Driving Bulbil Development of Pinellia ternata （Thunb.）Breit.PLOS ONE

Dear Dr. Du,

Thank you for submitting your manuscript to PLOS ONE. After careful consideration, we feel that it has merit but does not fully meet PLOS ONE’s publication criteria as it currently stands. Therefore, we invite you to submit a revised version of the manuscript that addresses the points raised during the review process.

We look forward to receiving your revised manuscript.

Kind regards,

Arun Kumar Shanker

Academic Editor

PLOS ONE

Journal Requirements:

Reviewers' comments:

Reviewer's Responses to Questions

**Comments to the Author**

1. If the authors have adequately addressed your comments raised in a previous round of review and you feel that this manuscript is now acceptable for publication, you may indicate that here to bypass the “Comments to the Author” section, enter your conflict of interest statement in the “Confidential to Editor” section, and submit your "Accept" recommendation.

Reviewer #1: All comments have been addressed

Reviewer #3: All comments have been addressed

2. Is the manuscript technically sound, and do the data support the conclusions?

Reviewer #1: Yes

Reviewer #3: Partly

3. Has the statistical analysis been performed appropriately and rigorously? 

Reviewer #1: Yes

Reviewer #3: Yes

4. Have the authors made all data underlying the findings in their manuscript fully available?

Reviewer #1: Yes

Reviewer #3: Yes

5. Is the manuscript presented in an intelligible fashion and written in standard English?

Reviewer #1: Yes

Reviewer #3: Yes

6. Review Comments to the Author

Reviewer #1: The manuscript is improved much. All the recommendations have been updated. However, would it be possible to add gene sequences within supplimentry 2 file. Yes you have mentioned ids of genes but hand on placement of gene sequences within manuscript material would make it more elegant

Reviewer #3: I would like to thank the authors for addressing the comments and suggestions thoroughly and thoughtfully. After reviewing the revised manuscript, I find that the changes made have significantly improved the quality and clarity of the study.

There are a few additional points that could be addressed to further enhance understanding:

The term "irregular observation" in the Materials section can be changed to "frequent observation" for better accuracy and clarity.

The quality of Figure 11 can be improved to ensure it is visually clear and easy to interpret.

Overall, the manuscript has improved considerably, and I commend the authors for their diligent efforts in revising the paper.

7. PLOS authors have the option to publish the peer review history of their article (what does this mean?). If published, this will include your full peer review and any attached files.

Reviewer #1: **Yes: **Dr. Shamshir ul Hussan

Reviewer #3: **Yes: **Vijayakumar Eswaramoorthy

---

## [Author Response · Author response to Decision Letter 1]

17 Oct 2024

[October 17th]

Emily Chenette

Editor-In-Chief

PLOS ONE

Dear Ph.D Arun Kumar Shanker: 

We wish to re-submit the manuscript titled “Transcriptomic and Biochemical Insights into Key Gene Networks Driving Bulbil Development of Pinellia ternata （Thunb.）Breit.” The manuscript ID is PONE-D-24-28450R1. 

Funding Statement: This work was supported by the Science and Technology Key R&D Program in Gansu Province (21YF5NA130) and the Special Foundation for Construction of National Traditional Chinese Medicine Industry Technology System in China "Supported by the earmarked fund for CARS-21".There was no additional external funding received for this study. The funders had no role in study design, data collection and analysis, decision to publish, or preparation of the manuscript

Based on your and the reviewers' suggestions, we have rechecked the manuscript and revised it appropriately. At the same time, we checked all references in the manuscript to make sure that it is complete and correct, and responded to and revised each of the review comments, as detailed below

Review Comments to the Author：

Reviewer #1:

1. The manuscript is improved much. All the recommendations have been updated. However, would it be possible to add gene sequences within supplimentry 2 file. Yes you have mentioned ids of genes but hand on placement of gene sequences within manuscript material would make it more elegant.

Authors’ Response: In response to your suggestion, we have added sequence information for the genes in Supplementary file 2.

Reviewer #3:

1. The term "irregular observation" in the Materials section can be changed to "frequent observation" for better accuracy and clarity.

Authors’ Response: We have changed the term “irregular observation” to “regular observation” in the revised manuscript.

2. The quality of Figure 11 can be improved to ensure it is visually clear and easy to interpret.

Authors’ Response: In response to your helpful suggestion, we have made appropriate adjustments to the Figure 11.

---

## [Decision Letter · Decision Letter 2]

11 Nov 2024

Transcriptomic and Biochemical Insights into Key Gene Networks Driving Bulbil Development of Pinellia ternata （Thunb.）Breit.

PONE-D-24-28450R2

Dear Dr. Du,

We’re pleased to inform you that your manuscript has been judged scientifically suitable for publication and will be formally accepted for publication once it meets all outstanding technical requirements.

Kind regards,

Arun Kumar Shanker

Academic Editor

PLOS ONE

Additional Editor Comments (optional):

Reviewers' comments:

Reviewer's Responses to Questions

**Comments to the Author**

1. If the authors have adequately addressed your comments raised in a previous round of review and you feel that this manuscript is now acceptable for publication, you may indicate that here to bypass the “Comments to the Author” section, enter your conflict of interest statement in the “Confidential to Editor” section, and submit your "Accept" recommendation.

Reviewer #1: All comments have been addressed

2. Is the manuscript technically sound, and do the data support the conclusions?

Reviewer #1: Yes

3. Has the statistical analysis been performed appropriately and rigorously? 

Reviewer #1: Yes

4. Have the authors made all data underlying the findings in their manuscript fully available?

Reviewer #1: Yes

5. Is the manuscript presented in an intelligible fashion and written in standard English?

Reviewer #1: Yes

6. Review Comments to the Author

Reviewer #1: We have read the manuscript and we believe that author has justifiably modified the manuscript and it might be acceptable for publication

7. PLOS authors have the option to publish the peer review history of their article (what does this mean?). If published, this will include your full peer review and any attached files.

Reviewer #1: **Yes: **SHAMSHIR UL HUSSAN

---

## [Editor Report · Acceptance letter]

18 Nov 2024

PONE-D-24-28450R2 

PLOS ONE

Dear Dr. Du, 

I'm pleased to inform you that your manuscript has been deemed suitable for publication in PLOS ONE. Congratulations! Your manuscript is now being handed over to our production team.

Kind regards, 

on behalf of

Dr. Arun Kumar Shanker 

Academic Editor

PLOS ONE